# AutoTool: Dynamic Tool Selection and Integration for Agentic Reasoning

**Jiaru Zou** [* 1 2]  **Ling Yang** [* 1 †]  **Yunzhe Qi** [2]  **Sirui Chen** [2]  **Mengting Ai** [2]  **Ke Shen**  **Jingrui He** [2 †]  **Mengdi Wang** [1 †]

Project Page: https://github.com/Gen-Verse/Open-AgentRL

## Abstract

Agentic reinforcement learning has advanced large language models (LLMs) to reason through long chain-of-thought trajectories while interleaving external tool use. Existing approaches assume a fixed inventory of tools, which limits the adaptability of LLM agents to new or evolving toolsets. We present **AutoTool**, a training framework that equips LLM agents with dynamic tool-selection capabilities throughout their reasoning trajectories. AutoTool employs a dual-phase optimization pipeline: (i) SFT and RL-based trajectory stabilization for coherent reasoning, and (ii) KL-regularized Plackett–Luce Ranking to refine consistent multi-step tool selection. We further build a 200k dataset with explicit tool-selection rationales across 1,000+ tools and 100+ tasks spanning mathematics, science, code generation, and multimodal reasoning. Across ten diverse benchmarks, we train two base models, Qwen3-8B and Qwen2.5-VL-7B, with AutoTool. With fewer parameters, AutoTool consistently outperforms advanced LLM agents and tool-integration methods, yielding average gains of 6.4% in math & science reasoning, 4.5% in search-based QA, 7.7% in code generation, and 6.9% in multimodal understanding. In addition, AutoTool exhibits stronger generalization by dynamically leveraging unseen tools from evolving toolsets during inference.

## 1. Introduction

Recent advances in reinforcement learning (RL) (Shao et al., 2024; Yu et al., 2025a) have improved the reasoning capabilities of large language models (LLMs) (Lu et al., 2025; Guo et al., 2025; Team et al., 2025), enabling them to generate long chain-of-thought (CoT) trajectories for complex tasks across domains such as visual question answering (Pramanick et al., 2025), knowledge retrieval (Su et al., 2025), and mathematical reasoning (Maxwell-Jia, 2024). Building on this, a growing line of research on agentic RL (Singh et al., 2025; Qian et al., 2025; Chen et al., 2025) explores how LLMs, beyond purely text-based model internal reasoning, can operate as agents that interleave their reasoning with multi-turn interactions in external tool environments (Mai et al., 2025; Wang et al., 2025c), such as search engines (Kong et al., 2023; Jin et al., 2025), code interpreters (Feng et al., 2025a), and vision tools (Su et al., 2025; Peng et al., 2025; Wu & Xie, 2024). The tool integration process reduces compounding errors in long CoT trajectories and enables more precise and reliable computation, which in turn strengthens the model's reasoning and leads to consistent task performance (Zhang et al., 2023; Feng et al., 2025b).

Despite these advances, existing approaches typically operate under the assumption of *single-domain fixed tools* (Gao et al., 2025a): the policy model learns when and how to invoke the designated tools, but the available tools are predefined and static for a specific task. In practice, these methods curate training datasets that explicitly cover the utilization of this fixed tool inventory, and then apply advanced supervised fine-tuning (SFT) or RL techniques to align the model's behavior with the prescribed tool usage (Feng et al., 2025a; Jin et al., 2025; Zou et al., 2026b). While effective within closed environments, such designs fail to capture realistic scenarios as illustrated in Figure 1(up), where (i) an agent must select the appropriate tool from a *complex domain-diverse toolset*, and (ii) *new unseen tools* during training stages may later be introduced and required at inference time. Without an explicit mechanism for dynamic tool selection, LLM agents risk overfitting to a closed tool inventory and failing to generalize in evolving tool environments. Addressing this gap requires LLM agents to move beyond simple tool-use toward adaptively choosing tools from dynamic toolsets during generation, while preserving strong reasoning ability. To this end, we aim to investigate:

> 💡 *How can LLM agents dynamically select tools during agentic reasoning?*

---

[*]Equal contribution  [1]Princeton University  [2]University of Illinois Urbana-Champaign.  Correspondence to: Ling Yang <ly1988@princeton.edu>, Jingrui He <jingrui@illinois.edu>, Mengdi Wang <mengdiw@princeton.edu>.

*Proceedings of the 43rd International Conference on Machine Learning*, Seoul, South Korea. PMLR 306, 2026. Copyright 2026 by the author(s).

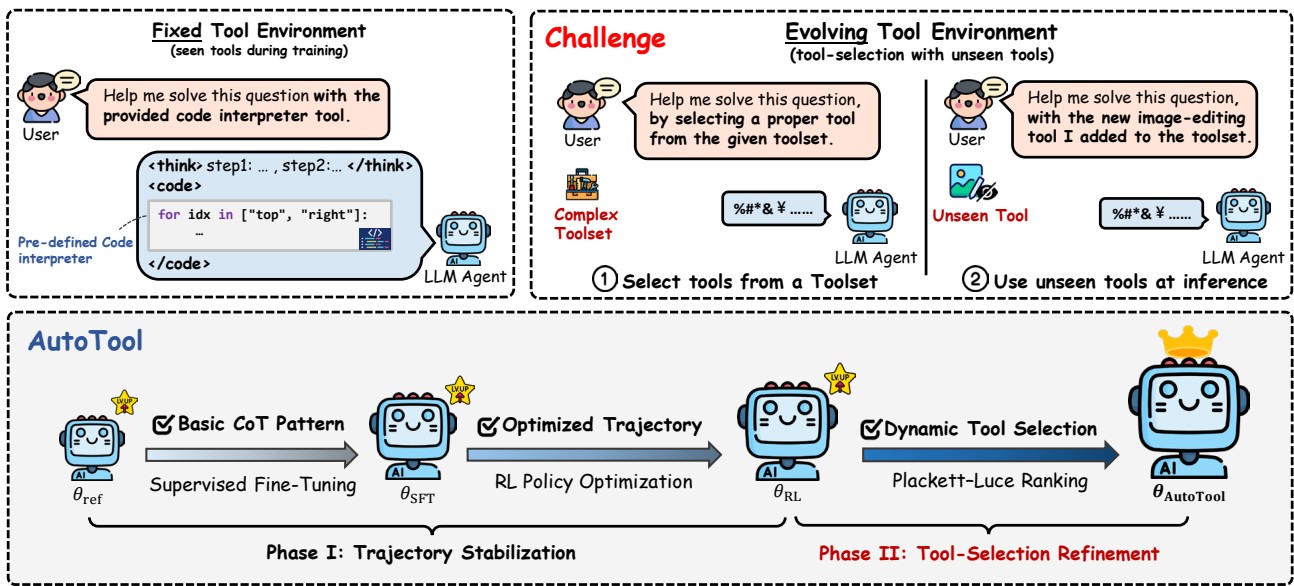

*Figure 1.* **Illustration of fixed vs. challenging evolving tool environments.** AutoTool enables LLM agents to dynamically select from complex unseen toolsets via a dual-phase optimization pipeline.

We introduce **AutoTool**, a new training framework that equips LLM agents with dynamic tool-selection capabilities throughout the reasoning process. AutoTool integrates reasoning and tool use into a unified trajectory, where the agent alternates between generating rationales and selecting tools from a large and evolving toolset. As illustrated in Figure 1 (bottom), we design a *dual-phase optimization pipeline* to train an LLM agent policy. In *Phase I*, we initialize the LLM agent with SFT followed by RL policy optimization, equipping the model with stable long CoT reasoning and trajectory generation abilities. In *Phase II*, we continue refining the LLM agent with a dedicated focus on tool selection, enabling the model to compose effective sequences of tool choices during its generation. We draw inspiration from the Plackett–Luce (PL) Ranking (Luce et al., 1959; Cheng et al., 2010) and frame tool-selection steps within model trajectories as a sequence ranking problem over the evolving toolset. We provide a theoretical bridge connecting the PL Ranking of tool-selection steps with the LLM agent's policy optimization, and then optimize the LLM agent with a KL-regularized objective by minimizing a tractable policy-level Cross-Entropy (CE) loss. The optimization over tool preferences explicitly trains the LLM agent to favor more effective tool compositions over weaker alternatives, leading to effective and generalizable tool-selection strategies.

To complement our method and support dynamic tool environments, we construct a *200k-scale dynamic tool-use dataset*, curated through a three-stage dedicated pipeline that explicitly incorporates tool-selection rationales into LLM agents' trajectory-level reasoning. Our curated dataset spans a diverse library of over 1,000 tools and more than 100 tasks across mathematics, science, code generation, and multimodal understanding, providing rich supervision for learning to choose tools from a large, evolving toolset.

We conduct extensive experiments across ten diverse benchmarks, training both Qwen3-8B (Yang et al., 2025a) and Qwen2.5-VL-7B (Bai et al., 2025) backbones with AutoTool. Incorporating AutoTool over previous training paradigms, such as SFT and GRPO, further boosts performance by an average of 6.4% in math & science, 4.5% in search, 7.7% in code generation, and 6.9% in multimodal understanding. Additionally, while advanced LLM agents or specialized tool-integration methods typically excel within fixed domains, they often fail to transfer their tool-usage abilities to other tasks. In contrast, AutoTool consistently demonstrates strong generalizability across all downstream tasks by leveraging dynamic tool-selection.

## 2. AutoTool

We introduce AutoTool, an agentic reasoning framework that equips LLM agents with dynamic tool-selection capabilities during generation. Our framework leverages a dual-phase optimization pipeline: (i) trajectory stabilization through supervised fine-tuning (SFT) and RL-based policy optimization, which endows agents with stable long-form CoT reasoning patterns; and (ii) tool-selection refinement via reward-guided Plackett–Luce (PL) ranking optimization.

**Method Roadmap.** We begin by introducing the preliminaries and notation that formalize the tool-selection setting within LLM agents' generation process (Section 2.1). We then describe how LLM agents interact with an evolving

toolset through tool-embedding grounded trajectory generation (Section 2.2). Next, we present the overall pipeline for the dual-phase policy optimization strategy (Section 2.3). Finally, we elaborate on how we optimize tool-selection via PL ranking during policy training (Section 2.4).

## 2.1. Preliminary and Notations

**Toolset.** We denote $\pi_\theta$ as an LLM agent, parameterized by $\theta$. In our agentic reasoning setting, an LLM agent is endowed with an *evolving toolset*, i.e., $T = \{t_1, t_2, \cdots, t_{|T|}\}$, where the size $|T|$ is not fixed. Each tool $t_k \in T$ is paired with a feature description $\mu(t_k)$, which serves as its "instruction manual," specifying the tool's name, functionality, input-output schema, and usage constraints.

**Agentic Trajectory Generation.** We conceptualize a complete reasoning trajectory generated by an LLM agent $\pi_\theta$ as an interplay among three complementary components:

$$\tau = \dots \tau_{i-1}^{\text{reason}} \oplus \tau_i^{\text{select}} \oplus \tau_{i+1}^{\text{integrate}} \dots, \quad (1)$$

where $\tau$ includes: (i) *inner-reasoning steps* $\tau^{reason}$, which $\pi_\theta$ produces a CoT format internal reasoning process; (ii) *tool-selection steps* $\tau^{select}$, which $\pi_\theta$ reasons over the toolset $T$ and selects a tool $t_k \in T$; and (iii) *tool-integration steps* $\tau^{integrate}$, which follow immediately after the tool-selection steps and involve calling the selected tool, executing it, and feeding the feedback back into $\pi_\theta$'s middle generation.

**Per-step Tool Selection.** During trajectory generation, we consider the tool-selection step $\tau_i^{\text{select}}$ in particular and formulate $\pi_\theta$'s tool-selection process as a *per-step decision-making problem*. Given the input question $x$, the evolving toolset $T$, and $\pi_\theta$'s previous generated steps $\tau_{<i}$, we model $\pi_\theta$'s generation at step $i$ as selecting $\tau_i^{\text{select}}$ from all potential tool-selection candidates $\{\tau_k^{\text{select}} \mid t_k \in T\}$ with each $\tau_k^{\text{select}}$ corresponds to the tool $t_k \in T$, i.e.,

$$\tau_i^{\text{select}} = \arg\max_{\tau_k^{\text{select}}} \pi_\theta\left(\tau_k^{\text{select}} \mid x, \tau_{<i}, T\right). \quad (2)$$

## 2.2. Tool-Aware Trajectory Generation

We first describe in AutoTool, how LLM agents interact with the external toolset $T$ and make tool selections during trajectory generation. Our rollout procedure is designed to allow LLM agents to seamlessly interleave reasoning with tool-selection steps for more coherent tool usage.

**Tool Embedding.** To incorporate rich and precise information from the external toolset $T$ into the LLM agent $\pi_\theta$, we first collect the embedding representation of each tool $t_k$ together with its descriptive features $\mu(t_k)$. We leverage the internal embedding layer of $\pi_\theta$ to obtain a contextualized representation for each tool:

$$\mathbf{e}_{t_k} = \text{Emb}_{\pi_\theta}[t_k, \mu(t_k)], \quad \text{with } \mathcal{E}_T = \{\mathbf{e}_{t_k}\}_{k=1}^{|T|}, \quad (3)$$

where $\mathcal{E}_T$ is the tool-embedding set for $T$. The latent embedding for each tool shares the same space of hidden state space as LLM agents, ensuring natural alignment with the LLM agents' generation process.

**Embedding-Anchored Tool Selection.** For each tool-selection step $\tau_i^{\text{select}}$, we ask $\pi_\theta$ to first generate a selection rationale $s_i$, followed by an explicit *anchor token* with predicted embedding representation $\mathbf{e}'_i$. The tool for $\tau_i^{\text{select}}$ is then selected by sampling from the softmax-normalized distance distribution (Dong et al., 2015) between the predicted $\mathbf{e}'_i$ and the candidate tool embeddings $\mathbf{e}_{t_k} \in \mathcal{E}_T$:

$$\pi_\theta(t_k \mid x, \tau_{<i}, s_i, T) = \frac{\exp\left(-\gamma \|\mathbf{e}'_i - \mathbf{e}_{t_k}\|_F^2\right)}{\sum_{t_j \in T} \exp\left(-\gamma \|\mathbf{e}'_i - \mathbf{e}_{t_j}\|_F^2\right)}, \quad (4)$$

where $||\cdot||_F$ is the Frobenius norm and $\gamma > 0$ controls the distribution skewness. By sampling from an embedding-based distribution, the LLM agent dynamically selects tools that are most aligned with its previous reasoning context. Note that Eq. 4's embedding-based selection operates over frozen, pre-computed tool embeddings, and the similarity computation involves only lightweight vector operations on a single hidden-state anchor. As a result, the tool-selection overhead is negligible relative to LLM inference and scales efficiently to toolsets with thousands of tools.

> **Takeaway: Why Embedding-Anchored Selection?**
>
> As the external toolset $T$ evolves, directly generating tool names by the LLM agent risks failure on unseen tools. By anchoring selection in the embedding space, which is generalizable (Mikolov et al., 2013; Ganguly et al., 2015), the agent can select new tools via *representation alignment*, avoiding reliance on memorized tool identifiers and remaining robust to the *evolving toolsets* environments.

After the tool selection step, the predicted tool is invoked and executed, with its outputs and feedback returned to the LLM agent for subsequent trajectory generation.

## 2.3. Dual-phase Policy Training Pipeline

Next, we describe how the LLM agent is trained as the policy model in AutoTool through a dual-phase optimization strategy. The overall training flow of $\pi_\theta$ is summarized as:

$$\underbrace{\theta_{\text{ref}} \xrightarrow{\text{SFT}} \theta_{\text{SFT}} \xrightarrow{\text{RL}}}_{\text{Phase I - Trajectory Stabilization}} \underbrace{\theta_{\text{RL}} \xrightarrow{\text{PL Ranking}} \theta_{\text{AutoTool}}}_{\text{Phase II - Tool-Selection Refinement}}$$

**Phase I – Trajectory Stabilization.** To enable the LLM agent to follow the reasoning patterns in Eq. 1 and produce valid tool-integrated trajectories, we initialize from a base reference model $\pi_{\theta_{\text{ref}}}$ (e.g., an off-the-shelf reasoning model). The LLM agent is then trained using agentic

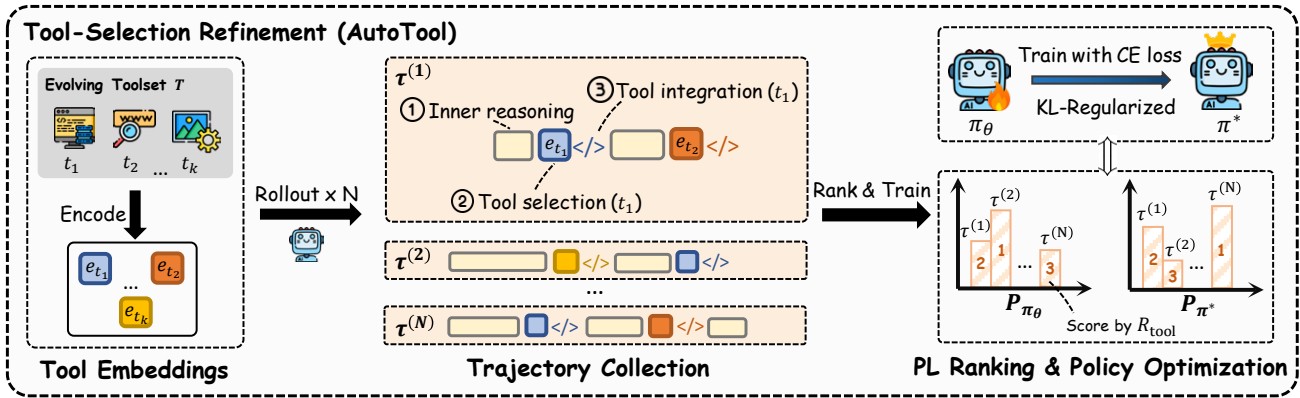

*Figure 2.* **Illustration on AutoTool's Tool-Selection Refinement Phase.** We cast tool-selection as a Plackett–Luce (PL) ranking problem during trajectory generation, optimizing the LLM agent to align its policy distribution with reward-consistent tool preferences.

training strategies (Dong et al., 2025; Feng et al., 2025a), beginning with supervised fine-tuning and followed by RL-based policy optimization. The primary goal of Phase I is to equip the LLM agent with stable generation across inner-reasoning, tool-selection, and tool-integration steps, which serves as the basis for the subsequent tool-selection refinement in Phase II.

**Phase II – Tool-Selection Refinement.** After the agent acquires stable reasoning capabilities in Phase I, we further refine the model with a dedicated optimization focused on tool-selection steps. Unlike the full-trajectory optimization in Phase I, we mask out internal reasoning and tool-integration steps, restricting optimization to the tool-selection segments of trajectories. This masked formulation places special emphasis on improving the tool-selection process.

**KL-Regularized Tool-Selection Objective.** In Phase II, we train the LLM agent via a KL-regularized RL objective to refine its tool-selection process. Formally, given an input question $x$ and the toolset $T$, our objective is to optimize the LLM agent policy from Phase I by:

$$
\begin{aligned}
\max_{\pi_\theta} \ & \mathbb{E}_{\tau \sim \pi_\theta(\cdot \mid x,T)}\big[R_{\text{tool}}(\tau)\big] \\
& - \beta \cdot D_{\text{KL}}\big(\pi_\theta(\cdot \mid x,T) \,\big\|\, \pi_{\text{old}}(\cdot \mid x,T)\big),
\end{aligned}
\tag{5}
$$

where $\beta$ controls the regularization intensity and $R_{\text{tool}}(\cdot)$ denotes the masked trajectory reward that evaluates solely on the correctness of tool-selection steps (defined later).

**Motivation on Policy Optimization under PL Ranking.** Previous agentic RL approaches (Shao et al., 2024; Yu et al., 2025a) directly optimize trajectory output rewards but do not explicitly capture the ordering relationships among candidate tools in the toolset. On the other hand, Plackett–Luce (PL) ranking (Luce et al., 1959; Qi et al., 2026) converts trajectory rewards into a permutation distribution, so that higher-reward tools are prioritized over weaker ones. This formulation naturally aligns with our embedding-anchored selection mechanism in Eq. 4. Motivated by this, we optimize Eq. 5 under the PL ranking framework, aligning the policy-induced ranking with reward-guided tool preferences and providing a principled refinement of tool-selection.

## 2.4. PL Ranking & Optimization on Tool Selection

We detail how the LLM agent is optimized under the PL ranking framework. The key idea is to view trajectory rollouts for each query as a ranked list, where the relative order is induced by trajectory rewards derived from tool-selection steps. By mapping these rewards into a probabilistic ranking distribution, we connect the LLM agent policy optimization with ranking consistency and train the policy with a tractable loss that directly improves the LLM agent's tool-selection.

**Trajectory Collection & Reward Assignment.** For each input question $x$ and toolset $T$, we first sample $N$ trajectory rollouts $\mathcal{T} = \{\tau^{(j)}\}_{j=1}^{N}$ from the LLM agent $\pi_\theta(\cdot \mid x,T)$. Within each trajectory, every tool-selection step $\tau_i^{\text{select}}$ is assigned a *step-level reward* $r_{\text{tool}}$ that jointly evaluates the quality of the tool-selection rationale and the correctness of the final answer produced with the chosen tool:

$$
r_{\text{tool}}(x, \tau_i^{\text{select}}) = \text{PRM}\big(\tau_i^{\text{select}}\big) + \text{Acc}(x, t_k), \quad t_k \in T,
\tag{6}
$$

where the process reward model, $\text{PRM}(\cdot)$, provides dense supervision by evaluating the reasoning quality of each tool-selection step $\tau_i^{\text{select}}$, and $\text{Acc}(x, t_k)$ measures whether the final answer obtained using tool $t_k$ is correct. We then aggregate all per-step selection rewards into a single masked *trajectory-level reward* $R_{\text{tool}}(\tau) = \frac{1}{|S(\tau)|} \sum_{i \in S(\tau)} r_{\text{tool}}(x, \tau_i^{\text{select}})$, where $S(\tau)$ is the (random) set of selection-step indices in each trajectory $\tau$. We use $R_{\text{tool}}$ for the KL-Regularized optimization in Eq. 5.

**PL Ranking over Collected Rollouts.** Given the collected set of trajectory rollouts $\mathcal{T}$ with associated tool-selection rewards $R_{\text{tool}}$, for a permutation $\sigma$ of $N$ trajectories, we

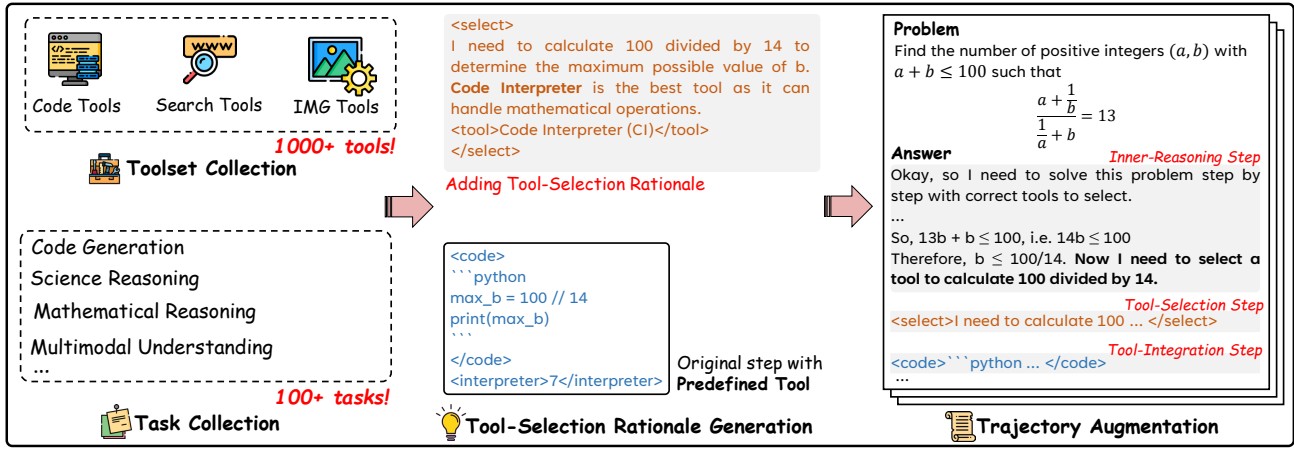

*Figure 3.* **Data curation pipeline for AutoTool** (Section 3). The overall pipeline has three stages: (i) *Toolset & Task Collection*, assembling 1,000+ tools with metadata across 100+ tasks in math, science, code, and multimodal reasoning; (ii) *Tool-Selection Rationale Generation*, producing explicit justifications for tool choices; and (iii) *Trajectory Augmentation*, combining CoT reasoning, tool-selection, and tool-integration steps into complete trajectories for robust training.

have the induced PL ranking model of the LLM agent $\pi_\theta$:

$$P_{\pi_\theta}(\sigma \mid \mathcal{T}) = \prod_{j=1}^{N} \frac{\exp\big(R_{\text{tool}}(\tau^{\sigma(j)})\big)}{\sum_{l=j}^{N} \exp\big(R_{\text{tool}}(\tau^{\sigma(l)})\big)}, \quad (7)$$

where $\sigma(j)$ denotes the trajectory placed at rank $j$ in the permutation $\sigma$. Eq. 7 shows that trajectory candidates with higher tool-selection reward appear earlier in the ranking order, thereby providing a tool-preference aware distribution over the $N$ candidate rollouts.

**Bridging PL Ranking with Policy Optimization.** In Eq. 7, directly optimizing the PL distribution over the $N!$ possible trajectory permutations is computationally infeasible (Cheng et al., 2010). To obtain a tractable training objective, we establish a theoretical bridge that links the PL ranking to the LLM agent's policy optimization. In the following theorem, we establish that learning the optimal policy is equivalent to matching its induced PL ranking distribution over candidate trajectory rollouts.

> **Theorem 2.1** (**Equivalence between Optimal Policy and PL Ranking**). *Consider the KL-regularized RL objective defined in Eq. 5 with the tool-selection reward $R_{tool}$ defined in Eq. 6. Let $\pi^*$ denote the corresponding optimal policy (Rafailov et al., 2023) for Eq. 5. Then, a trainable policy $\pi_\theta$ is equal to the optimal policy (i.e., $\pi_\theta = \pi^*$) if and only if their induced PL ranking distributions coincide for any input $x$ and trajectory collection $\mathcal{T}$, i.e.*
>
> $$\pi_\theta = \pi^* \iff P_{\pi_\theta}(\sigma \mid \mathcal{T}) = P_{\pi^*}(\sigma \mid \mathcal{T}), \quad \forall \sigma.$$

Theorem 2.1 (proof in Appendix B) suggests that optimizing the LLM agent policy provides a tractable surrogate to direct PL ranking optimization. Accordingly, leveraging the closed-form solution of $\pi^*$ (Schulman et al., 2017; Rafailov et al., 2023), we employ a practical Cross-Entropy (CE) loss

to train the LLM agent policy $\pi_\theta$ towards the optimal policy $\pi^*$ on a training set $\mathcal{D}$:

$$\mathcal{L}_{\text{CE}} = -\mathbb{E}_{x \sim \mathcal{D}} \sum_{\tau \in \mathcal{T}} \pi^*(\tau \mid x, \mathcal{T}) \log \pi_\theta(\tau \mid x, \mathcal{T})$$

$$= -\mathbb{E}_{x \sim \mathcal{D}} \sum_{\tau \in \mathcal{T}} \frac{\pi_{\text{old}}(\tau \mid x, T) \exp\big(\frac{1}{\beta} R_{\text{tool}}(\tau)\big)}{\sum_{\tau' \in \mathcal{T}} \pi_{\text{old}}(\tau' \mid x, T) \exp\big(\frac{1}{\beta} R_{\text{tool}}(\tau')\big)}$$

$$\log \frac{\pi_\theta(\tau \mid x, T)}{\sum_{\tau' \in \mathcal{T}} \pi_\theta(\tau' \mid x, T)}. \quad (8)$$

## 3. Data Curation for Tool-Selection

Since existing training data (Yu et al., 2025a; Feng et al., 2025a) and downstream evaluations (Qian et al., 2025; Wang et al., 2025a) for LLM agents prescribe fixed tool usage tailored to specific tasks, they cannot be directly applied to train or evaluate tool selection setups. To address this limitation, we extend these data recipes to curate a 200k dataset that *explicitly integrates tool selection into the reasoning trajectory*. In particular, we design a data curation pipeline that simulates real-world scenarios of agentic tool selection and reasoning across diverse domains, including mathematical and scientific reasoning, code generation, and multimodal understanding. As illustrated in Figure 3, our pipeline consists of three key stages:

❶ **Toolset & Task Collection.** We begin by assembling a diverse library of candidate tools, drawing from prior tool integration studies (Su et al., 2025; Feng et al., 2025a; Jin et al., 2025). In particular, we collect three representative categories: (i) *Code Tools*, including code sandboxes and interpreters; (ii) *Search Tools*, covering search engines and web browser APIs (e.g., Jina Reader); and (iii) *Multimodal Tools*, comprising image processing and understanding mod-

ules such as OCR (Su et al., 2025) and GroundingDINO (Liu et al., 2024). We collect the total of 1,346 tools during this process. Next, we correspondingly curate diverse agentic tasks aligned with these tools, spanning mathematical and scientific reasoning, code generation, search-based QA, and related domains, covering 120 task types. Together, the toolset and task set provide the foundation for enabling agents to operate in open-ended, multimodal environments.

❷ **Tool-Selection Rationale Generation.** As real-world toolsets are dynamic and may include unseen tools, Auto-Tool (Section 2) formulates tool selection as a step-wise decision-making process in which the agent reasons toward selecting the most appropriate tool given the current context. To support this from a data perspective, we revisit each collected trajectory and, before every tool invocation, prompt an expert reasoning model to generate an explicit rationale explaining why the selected tool is suitable at that step. The rationales provide supervision that aligns tool selection with contextual reasoning rather than fixed tool identities.

❸ **Trajectory Augmentation.** Next, we filter and insert newly generated tool-selection rationales back into the original trajectories. To ensure data quality, we first apply LLM-as-a-judge (Zheng et al., 2023) to filter out rationales that result in incorrect tool choices. The remaining valid rationales are placed between the model's internal reasoning and the subsequent tool invocation (Figure 3). Finally, we use the expert model to review and refine the full trajectories, smoothing transitions, and removing logical inconsistencies. This process produces a corpus of 200k high-quality trajectories with explicit reasoning, tool-selection, and tool-integration steps (Eq. 1). Additional dataset statistics and dynamic tool-selection settings are provided in Appendix A.1.

**Supporting Unseen and Evolving Toolsets.** To support the setting of unseen and evolving tool environments, we explicitly decouple toolset construction from training-instance curation and progressively expand the tool-selection pool during inference. Specifically, among the 1,346 tools constructed in Stage ❶ , only a subset of 460 tools (34.2%) aligned with the training tasks is used during AutoTool 's dual-stage training. For the rationales curated in Stages ❷ and ❸ , all training instances involve tool-selection rationales drawn exclusively from the 460-tool subset. During downstream inference, we progressively expand the tool-selection pool across multiple rounds, introducing additional tools beyond those seen during training. In the final training stage, the full tool library of 1,346 tools is available, of which 886 tools (65.8%) are entirely unseen during training. This separation between toolset construction and training-instance curation enables AutoTool to operate under evolving tool environments and to fairly evaluate its ability to select previously unseen tools at inference. Additional data statistics and settings are listed in Appendix A.2 and A.3.

## 4. Experiments

**Tasks and Datasets.** To assess the effectiveness and generalizability of AutoTool, we conduct evaluations on a broad range of downstream tasks. For *mathematical and scientific reasoning*, we evaluate on AIME24 (Maxwell-Jia, 2024), AIME25 (math ai, 2025), and GPQA-Diamond (Rein et al., 2024). For *search-based reasoning*, we use multi-hop question answering datasets including HotpotQA (Yang et al., 2018), 2WikiMultiHopQA (2Wiki) (Ho et al., 2020), and Bamboogle (Press et al., 2023), which together cover a broad spectrum of search and reasoning challenges. For *multimodal understanding*, we first apply MMSearch (Jiang et al., 2024), a comprehensive benchmark for evaluating multimodal search performance. In addition, to assess AutoTool performance in multi-turn interaction scenarios, we curate 500 image-based questions spanning three domains: (i) V-Chart, covering chart reasoning with questions sourced from the ChartGemma dataset (Masry et al., 2024); (ii) V-Math, consisting of math problems derived from GSM8K (Cobbe et al., 2021) and AIME24 (Maxwell-Jia, 2024); and (iii) V-Code, focusing on code generation with questions drawn from MBPP (Austin et al., 2021), HumanEval (Chen et al., 2021), and LiveCodeBench (Jain et al., 2024). All three domains are presented to the LLM agents in image form, requiring them to parse visual inputs before reasoning.

**Baselines and Models.** We compare AutoTool with a broad set of baselines, including advanced LLM agents, fixed tool-integration and selection approaches, and agentic training paradigms. (i) For *advanced LLM agents*, we compare against strong reasoning models including GPT-4o (Hurst et al., 2024), Qwen2.5-VL-72B (Bai et al., 2025), Qwen2.5-Math-72B (Yang et al., 2025a), QwQ-32B (Team, 2025), and DeepSeek-R1-Distill-Qwen-14B (Guo et al., 2025). (ii) For *tool-integration methods*, we compare with both general and domain-specific methods: Toolformer (Schick et al., 2023) and ToolGen (Wang et al., 2025b) as general tool-integration; ReTool (Feng et al., 2025a) for strategic tool use on math and science reasoning, Search-R1 (Jin et al., 2025) for search-based question answering, and v-ToolRL (Su et al., 2025) for multimodal understanding. (iii) For *agentic training methods*, we compare with three common agentic post-training paradigms: supervised fine-tuning (SFT), where the policy model is fine-tuned on our curated training set; Agentic RL algorithms, including GRPO (Shao et al., 2024) and ARPO (Lu et al., 2025).

**Implementation Details.** We incorporate AutoTool into two off-the-shelf models Qwen3-8B (think mode) (Yang et al., 2025a) and Qwen2.5-VL-7B (Bai et al., 2025), and train them on our curated training dataset described in Section 3. For Phase I training, we leverage LLaMa-Factory (Zheng et al., 2024) for SFT and VeRL (Sheng et al., 2024) for RL training. For Phase II training, we train the induced

*Table 1.* **Comparison of AutoTool with advanced LLM agents and tool-integration methods across math, search, and multimodal benchmarks.** While some baselines excel in isolated domains (e.g., ReTool on math, Search-R1 on search, GPT-4o on multimodal), AutoTool delivers balanced and robust performance across all diverse tasks, demonstrating the generalization benefits of dynamic tool-selection and integration. The best results are **bold**, and the second-best results are underlined.

| Method | Size | Math ↑ | | Search ↑ | | Multimodal ↑ | | |
| --- | --- | --- | --- | --- | --- | --- | --- | --- |
| | | AIME24 | AIME25 | HotpotQA | 2Wiki | V-Chart | V-Math | V-Code |
| *Advanced LLM Agents* | | | | | | | | |
| GPT4o | – | 18.2 | 15.8 | 38.7 | 29.8 | 27.3 | 41.4 | 51.2 |
| Qwen2.5-VL-72B-Instruct | 72B | 6.3 | 4.1 | 13.9 | 10.4 | 22.5 | 24.5 | 18.2 |
| Qwen2.5-Math-72B-Instruct | 72B | 31.3 | 27.4 | 23.7 | 16.7 | – | – | – |
| QwQ-32B | 32B | 47.3 | 31.8 | 32.6 | 22.3 | – | – | – |
| DeepSeek-R1-Distill-Qwen-14B | 14B | 65.9 | 44.1 | 24.2 | 14.1 | – | – | – |
| *Fixed Tool-Integration Methods* | | | | | | | | |
| Toolformer | 6B | 6.4 | 4.8 | 11.5 | 10.6 | – | – | – |
| ToolGen | 8B | 14.3 | 12.6 | 19.8 | 17.5 | – | – | – |
| v-ToolRL | 2B | – | – | – | – | 21.6 | 19.5 | 13.3 |
| Search-R1 | 7B | 12.1 | 7.3 | 43.3 | 44.5 | – | – | – |
| ReTool | 32B | **70.1** | 47.4 | 21.4 | 19.7 | – | – | – |
| *Dynamic Tool-Selection & Integration* | | | | | | | | |
| **AutoTool (Qwen2.5-VL-7B)** | 7B | 45.3 | 38.9 | 33.2 | 36.5 | 13.2 | 44.3 | 52.5 |
| **AutoTool (Qwen3-8B)** | 8B | 68.8 | **51.2** | **45.1** | **48.8** | **24.7** | **53.0** | **56.1** |

PL-Ranking framework by setting the rollout size $N$ to 8 per question and training for 3 epochs. All training and inference experiments are conducted on 8xA100-80G GPUs. We leave additional experimental setups in Appendix C.

### 4.1. Main Results

**AutoTool Balances Performance Across Domains.** As shown in Table 1, we first compare AutoTool against advanced LLM agents and tool-integration methods. Auto-Tool consistently delivers balanced and robust performance across diverse domains. Large LLM agents such as GPT-4o and Qwen2.5-VL-72B rely primarily on internal reasoning and exhibit uneven performance across tasks (e.g., GPT-4o excels in multimodal benchmarks but underperforms on math-intensive tasks, while Qwen2.5-Math-72B shows the opposite trend). Fixed tool-integration methods similarly suffer from domain brittleness. ReTool achieves strong math performance but degrades substantially on search tasks. In contrast, by dynamically selecting and integrating tools during reasoning, AutoTool extends the model's effective capability beyond its internal knowledge and achieves consistently strong results across various benchmarks.

**AutoTool Advances beyond Standard Agentic Training.** Table 2 compares AutoTool with two commonly adopted agent training paradigms. For Qwen3-8B, we equipped the model with vision-extraction tools such as OCR to retrieve text from image inputs. AutoTool outperforms both SFT

and RL methods by explicitly modeling and optimizing tool selection within reasoning trajectories. For instance, Auto-Tool (Qwen3-8B) delivers an average of 6.7% improvement on AIME24 and 6.5% on HotpotQA compared with GRPO and ARPO. Our results demonstrate that, beyond stabilizing reasoning via Phase I, AutoTool's Phase II PL-ranking provides additional benefits by explicitly modeling dynamic tool selection, thereby enabling tool-aware optimization.

### 4.2. In-depth Analyses on AutoTool

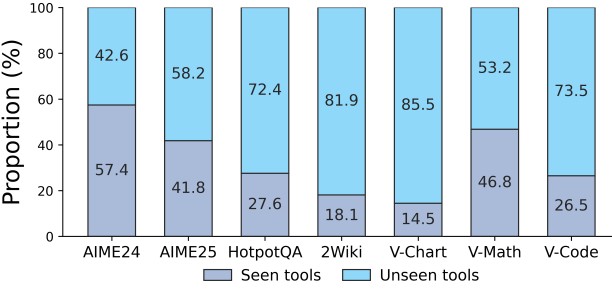

*Figure 4.* Proportion of seen vs. unseen tools selected by AutoTool (Qwen3-8B) across 7 benchmarks.

**Generalization on Unseen Tools.** To better understand AutoTool's selection ability on unseen tools, we analyze AutoTool (Qwen3-8B) generated trajectories in Table 1. We extract all tool-selection steps and identify the selected tools. Across all 7 tasks, we compute the fractions of *seen v.s. unseen* tools among runs that lead to correct final answers. As shown in Figure 4, AutoTool relies largely on unseen tools,

*Table 2.* **Comparison of AutoTool with agentic training baselines.** AutoTool consistently outperforms the training-based baselines, demonstrating that the Phase II PL-ranking stage provides complementary gains by explicitly optimizing dynamic tool selection.

| Method | Math & Science ↑ | | | Search ↑ | | | Multimodal ↑ | | | |
|---|---|---|---|---|---|---|---|---|---|---|
| | AIME24 | AIME25 | GPQA | HotpotQA | 2Wiki | Bamboogle | V-Chart | V-Math | V-Code | MMSearch |
| **Qwen3-8B** | 46.7 | 40.0 | 58.1 | 26.2 | 31.1 | 37.4 | 6.0 | 39.4 | 43.2 | – |
| SFT | 53.3 | 40.0 | 63.6 | 34.3 | 37.9 | 40.2 | 16.3 | 44.6 | 48.6 | – |
| GRPO | 56.7 | 46.7 | 70.7 | 38.6 | 45.2 | 52.8 | 22.8 | 51.8 | 55.2 | – |
| ARPO | 66.7 | 50.0 | 72.8 | 43.6 | 48.1 | 54.5 | 23.2 | 52.6 | 55.8 | – |
| **AutoTool** | **66.7** | **53.3** | **73.7** | **45.1** | **48.8** | **56.8** | **24.7** | **53.0** | **56.1** | – |
| **Qwen2.5-VL-7B** | 6.7 | 13.2 | 7.6 | 14.0 | 18.2 | 10.6 | 2.6 | 22.9 | 26.6 | 27.8 |
| SFT | 30.0 | 20.0 | 30.8 | 24.9 | 22.0 | 24.5 | 6.8 | 28.4 | 43.2 | 39.7 |
| GRPO | 36.7 | 33.3 | 33.3 | 28.5 | 32.5 | 44.0 | 10.7 | 42.7 | 49.6 | 45.2 |
| ARPO | 46.7 | **43.3** | 36.1 | 32.0 | 35.1 | 46.4 | 13.2 | 43.0 | 50.9 | 48.2 |
| **AutoTool** | **46.7** | 40.0 | **38.4** | **33.2** | **36.5** | **48.2** | 13.2 | **44.3** | **52.5** | **49.3** |

*Table 3.* Out-of-domain performance under fully unseen tool settings. AutoTool is evaluated using only unseen tools at inference.

| Method | AIME24 | HotpotQA | LiveCodeBench | MedQA |
|---|---|---|---|---|
| Search-R1 | 13.3 | 43.3 | 36.2 | 73.5 |
| ReTool | 66.7 | 21.4 | 71.5 | 64.7 |
| **AutoTool (with unseen tools)** | **66.7** | **43.8** | **77.9** | **84.3** |

especially on search and multimodal tasks. This suggests that the improved performance is driven by generalizing tool selection beyond memorizing training-time tools.

**Out-of-Domain Performance under Unseen Tools.** We further evaluate AutoTool in a fully unseen-tool setting by restricting the inference-time tool pool to the 886 unseen tools and testing on AIME24 and V-Chart. We additionally assess out-of-domain generalization on two benchmarks not covered by our curated dataset, LiveCodeBench-V2 (Jain et al., 2024) and MedQA (Yang et al., 2025b). All experiments are conducted using AutoTool (Qwen3-8B) under identical inference settings. As shown in Table 3, AutoTool consistently achieves the strongest performance across all evaluated domains. In addition, relative to using the full toolset (1,346 tools), restricting AutoTool to unseen tools results in only a marginal performance drop of less than 2% on both AIME24 and HotpotQA.

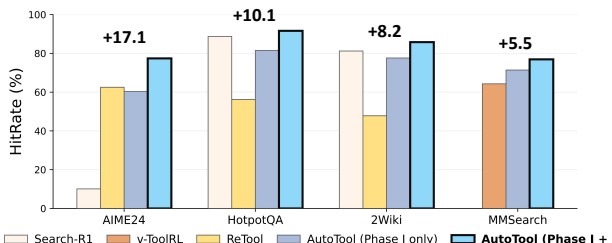

*Figure 5.* Analyses on Tool-selection hit rates. Higher hit rates correlate with improved downstream performance.

**Relationship between Tool-Selection and Performance.** To further examine the relationship between tool-selection capability and final performance, we analyze the overall *HitRate* of AutoTool. Comparing Figure 5 with the main results in Tables 1 and 2, we find that higher tool-selection hit rates consistently correlate with improved downstream accuracy across methods. Notably, Phase-II refinement substantially increases AutoTool 's HitRate, which directly translates into stronger end-task performance.

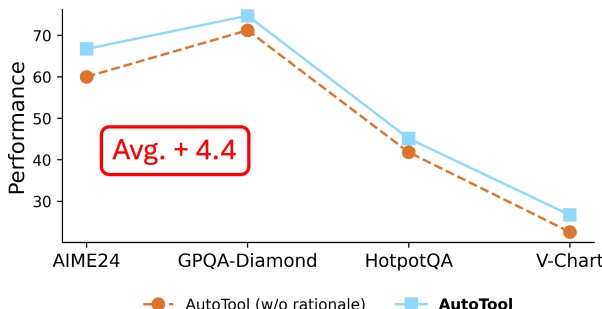

*Figure 6.* Ablations on tool-selection rationales.

**Effectiveness of Tool-Selection Rationales.** We conduct an ablation study to evaluate the effectiveness of incorporating tool-selection rationales. As shown in Figure 6, models trained with rationales consistently outperform those without. In addition, we observe that tool-selection rationales increase successful tool executions by 13.5%, indicating more reliable tool use during intermediate steps. We leave additional analyses and case studies in Appendix E.

## 5. Related Work

**Agentic Reasoning in RL.** Recent work on agentic reasoning in LLMs leverages reinforcement learning to directly optimize reasoning behaviors. The introduction of critic-free methods such as GRPO and its variants (Li et al., 2025a;

Yu et al., 2025a; Zou et al., 2026c) marked a key shift toward group-relative advantage estimation. Building on this, subsequent studies incorporate tool use into agentic RL, including ARTIST (Singh et al., 2025) and rStar2 (Shang et al., 2025), while parallel lines of work on self-evolving agents emphasize continual adaptation through learning and structural modification (Fang et al., 2025).

**Tool Use and Integration in LLMs.** Early work on tool use in LLMs relied on prompting strategies that interleaved reasoning with external queries or API calls (Press et al., 2023; Yao et al., 2023; Zou et al., 2026a). Additional existing works (Nakano et al., 2021; Schick et al., 2023; Hu et al., 2024) proposed training-based approaches that explicitly empower LLMs with tool-use capabilities, as well as extended tool-usage to large-scale API integration and planning (Patil et al., 2024; Song et al., 2023), multi-tool orchestration (Shen et al., 2023), and adaptive selection (Qin et al., 2024). More recent work integrates tool use tightly into training and system design, such as multimodal agent tuning (Gao et al., 2025b) and infrastructure for dynamic discovery and synchronization (Ding, 2025; Mastouri et al., 2026). We leave additional related works in Appendix D.

## 6. Conclusion

We introduced AutoTool, a framework that equips LLM agents with dynamic tool-selection capabilities. We train the agent policy through a dual-phase optimization pipeline, enabling AutoTool to move beyond static tool use and achieve adaptive, generalizable reasoning. Extensive experiments across mathematics, science, retrieval, and multimodal tasks demonstrate that AutoTool consistently outperforms training-only baselines and existing tool-enhanced methods. These results highlight the importance of explicit tool-selection optimization in building extensible agents that can effectively operate under evolving tool environments.

## Acknowledgment

The authors thank members of the iDEA-iSAIL Group for helpful discussions and feedback on this work. This work is supported by National Science Foundation under Award No. IIS-2117902. The views and conclusions are those of the authors and should not be interpreted as representing the official policies of the funding agencies or the government.

## Impact Statement

AutoTool operates within standard large language model deployment settings and focuses on optimizing inference-time decision-making over existing tool interfaces. It does not introduce new forms of autonomy or human replacement beyond current agentic systems. We don't anticipate potential negative societal impacts arising uniquely from this work.

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

# Table of Contents

# A. Additional Details on AutoTool Data Curation

## A.1. Data Curation Pipeline Details

**❶ Toolset & Task Collection.** We begin by assembling a diverse library of candidate tools, drawing from prior tool integration studies (Su et al., 2025; Feng et al., 2025a; Jin et al., 2025). In particular, we collect three representative categories: (i) *Code Tools*, including code sandboxes and interpreters that allow the agent model to execute code and leverage outputs or error messages during generation; (ii) *Search Tools*, covering search engines and web browser APIs (e.g., Jina Reader) for retrieving external knowledge from various corpora; and (iii) *Image Tools*, comprising image processing and understanding modules such as OCR (Su et al., 2025) and GroundingDINO (Liu et al., 2024), which enable the agent model to extract and reason over textual content embedded in images. This results in a comprehensive toolset of over 1,000 tools. Each tool is annotated with a feature description specifying its toolset index, functionality, input-output schema, and usage constraints. In addition, we curate a diverse set of downstream tasks aligned with these tools, spanning mathematical and scientific reasoning, code generation, search-based QA, and related domains, covering over 100 task types. Together, the toolset and task set provide the foundation for enabling agents to operate in open-ended, multimodal environments.

**❷ Tool-Selection Rationale Generation.** In real-world scenarios, toolsets are dynamic and evolve with new incoming tools. Therefore, we avoid formulating tool selection as a simple classification problem over a fixed tool index. Instead, we cast it as a decision-making process in which the agent model should learn to reason step by step toward selecting the most appropriate tool for the current context. To support this, we first retrieve the original responses from the collected downstream tasks in the previous stage, which consist of reasoning trajectories involving both model reasoning and tool integration components. In each response chain, before a specific tool is invoked, we leverage an expert reasoning model (DeepSeek-R1) to generate explicit rationales that justify why a particular tool is preferred at each step, serving as the supervision signals that align tool selection with contextual reasoning. We illustrate our template to generate tool-selection rationale training data in Figure 7.

**❸ Trajectory Augmentation.** After collecting tool-selection rationales, we insert them back into the original response trajectories. To ensure quality, we first leverage LLM-as-a-judge (Zheng et al., 2023) to filter out invalid rationales that lead to incorrect tool choices. We then insert the valid rationales between the model's internal reasoning and the subsequent tool invocation (see the example in Figure 3). Finally, we employ DeepSeek-R1 to review the entire trajectory, smoothing connections and eliminating logical inconsistencies. This process yields a corpus of 200k data instances, including various tasks with corresponding tool selection and integration trajectory responses.

## A.2. Data Curation Statistics

Through Stage 1: Toolset & Task Collection, we curate a comprehensive toolset of 1,346 tools drawn from diverse domains (e.g., online APIs, open-source libraries, multimodal utilities). Among them, 460 tools are used during training, while the remaining 886 tools are kept completely unseen and reserved exclusively for inference-time evaluation. During training, the candidate toolset is fixed to the 460 predefined tools, because all tool-selection rationales and ground-truth trajectories in our source datasets involve only these tools. This fixed training pool ensures stable tool-embedding learning and consistent PL-ranking optimization. Note that Stage 1 is decoupled from Stages 2 and 3, which focus on collecting high-quality reasoning trajectories strictly over the 460 seen tools. This separation prevents leakage of unseen tools into training and enables a clean evaluation of generalization.

During inference, we expand the available toolset to the full 1,346 tools, simulating an *evolving* tool environment in which many tools have never been observed during training. This design allows us to systematically evaluate AutoTool's ability to perform dynamic tool selection under distribution shift, where the candidate tools may differ across tasks and include entirely new unseen tools. The overall dataset statistics are summarized in Table 4. We next describe how experiments are conducted under the evolving toolset setting.

*Table 4.* Statistics of the curated toolset.

| Category | #Tools |
|---|---|
| Full Toolset | 1346 |
| Seen Tools (Training) | 460 |
| Unseen Tools (Inference Only) | 886 |

## A.3. Evolving Toolset Settings

**Training-Time Toolset.** During training, the toolset is fixed to the 460 "seen" tools collected through Stage 2 and Stage 3 of our data curation pipeline. All training-time tool-selection rationales and ground-truth tool trajectories reference only

these 460 tools because the underlying source datasets provide demonstrations exclusively for this subset. As a result, every training instance uses the same 460-tool candidate pool. This design ensures stable tool-embedding learning and consistent PL-ranking optimization across all training samples. As AutoTool learns dynamic tool-selection behavior through its embedding-based matching mechanism (Eq. 4), rather than through dynamic changes in the training-time toolset itself. Therefore, an evolving toolset is not required during training for the model to develop dynamic selection capabilities.

**Inference-Time Evolving Toolset.** During inference, AutoTool does not assume a static or closed tool inventory. Instead, we simulate realistic tool evolution by evaluating the model over the full toolset that includes both the 460 tools seen during training and an additional 886 unseen tools from our toolset collection. This allows us to evaluate AutoTool under realistic, large-scale tool environments while keeping comparisons fair across baselines.

### A.4. Template on Generating Tool-Selection Rationales

---

**Template for Tool-Selection Rationale**

You are a Tool Use Expert in selecting the most appropriate tool(s) to answer user questions. You are provided with three inputs:

# INPUT QUESTION: ...

# MODEL ANSWER: ...

# TOOLSET (including different types of tools and capabilities)
¡tool1 name¿ + ¡description¿
¡tool2 name¿ + ¡description¿
...
Your task is to provide the rationale and the final selected tool from the toolset before the model invokes a tool. Your response:

---

*Figure 7.* Template to generate tool-selection rationales during AutoTool's data curation process.

# B. Theoretical Bridge between Optimal Policy and PL Ranking

### B.1. Proof of Theorem 2.1

In the main body, we denote $P_\pi(\sigma|\mathcal{T})$ by omitting the query $x$ for notation brevity. For the derivations below, we reinstate this dependence. In this case, for a collection of trajectories $\mathcal{T} = \{\tau^{(j)}\}_{j=1}^{|\mathcal{T}|}$ and query $x$, the Plackett-Luce (PL) ranking model induced by a policy $\pi$, relative to a previous policy $\pi_{\text{old}}$, will be defined as:

$$P_\pi(\sigma|\mathcal{T}, x) = \prod_{i=1}^{|\mathcal{T}|} \frac{\exp\left(\beta \log \frac{\pi(\tau^{\sigma(i)}|x)}{\pi_{\text{old}}(\tau^{\sigma(i)}|x)}\right)}{\sum_{j=i}^{|\mathcal{T}|} \exp\left(\beta \log \frac{\pi(\tau^{\sigma(j)}|x)}{\pi_{\text{old}}(\tau^{\sigma(j)}|x)}\right)}, \tag{9}$$

where we have $\sigma$ being a permutation (or ranking) of the indices $\{1, 2, \ldots, |\mathcal{T}|\}$, and denote $\tau^{\sigma(i)}$ being the trajectory ranked at position $i$ in permutation $\sigma$. $\frac{\pi(\tau|x)}{\pi_{\text{old}}(\tau|x)}$ represents the relative preference of policy $\pi$ over the previous checkpoint before updating.

We consider two policy models: $\pi_\theta$ is a trainable policy model with parameters $\theta$, and $\pi^*$ is the optimal policy model. Subsequently, let us denote the two induced PL ranking models by

$$P_{\pi_\theta}(\sigma|\mathcal{T}, x) = \prod_{i=1}^{|\mathcal{T}|} \frac{\exp\left(\beta \log \frac{\pi_\theta(\tau^{\sigma(i)}|x)}{\pi_{\text{old}}(\tau^{\sigma(i)}|x)}\right)}{\sum_{j=i}^{|\mathcal{T}|} \exp\left(\beta \log \frac{\pi_\theta(\tau^{\sigma(j)}|x)}{\pi_{\text{old}}(\tau^{\sigma(j)}|x)}\right)},$$

$$P_{\pi^*}(\sigma|\mathcal{T}, x) = \prod_{i=1}^{|\mathcal{T}|} \frac{\exp\left(\beta \log \frac{\pi^*(\tau^{\sigma(i)}|x)}{\pi_{\text{old}}(\tau^{\sigma(i)}|x)}\right)}{\sum_{j=i}^{|\mathcal{T}|} \exp\left(\beta \log \frac{\pi^*(\tau^{\sigma(j)}|x)}{\pi_{\text{old}}(\tau^{\sigma(j)}|x)}\right)}.$$

We will then have the following result on the equivalence between the optimal policy and the optimal PL ranking model. Results from the following Theorem B.1 supports the conclusion from Theorem 2.1.

> **Theorem B.1** (**Equivalence between Optimal Policy and PL Ranking**). *Consider the KL-regularized RL objective defined in Eq. 5 with the tool-selection reward $R_{tool}$ defined in Eq. 6. Let $\pi^*$ denote the corresponding optimal policy (Rafailov et al., 2023) for Eq. 5. Then, a trainable policy $\pi_\theta$ is equal to the optimal policy (i.e., $\pi_\theta = \pi^*$) if and only if their induced PL ranking distributions coincide for any input $x$ and trajectory collection $\mathcal{T}$, i.e.,*
>
> $$\pi_\theta = \pi^* \iff P_{\pi_\theta}(\sigma \mid \mathcal{T}) = P_{\pi^*}(\sigma \mid \mathcal{T}), \quad \forall \sigma.$$

*Proof.* We need to prove both directions of the equivalence, and we will start with the forward direction that optimal policy indicates the optimal PL ranking.

**Forward Direction.** Suppose $\pi_\theta(\tau|x) = \pi^*(\tau|x)$ for all $\tau$ and $x$. Then, since the policies are identical, their ratios with respect to the old policy are also identical, leading to

$$\frac{\pi_\theta(\tau|x)}{\pi_{\text{old}}(\tau|x)} = \frac{\pi^*(\tau|x)}{\pi_{\text{old}}(\tau|x)} \tag{10}$$

Therefore, with $\beta > 0$, for any permutation $\sigma$:

$$P_{\pi_\theta}(\sigma|\mathcal{T}, x) = \prod_{i=1}^{|\mathcal{T}|} \frac{\exp\left(\beta \log \frac{\pi_\theta(\tau^{\sigma(i)}|x)}{\pi_{\text{old}}(\tau^{\sigma(i)}|x)}\right)}{\sum_{j=i}^{|\mathcal{T}|} \exp\left(\beta \log \frac{\pi_\theta(\tau^{\sigma(j)}|x)}{\pi_{\text{old}}(\tau^{\sigma(j)}|x)}\right)} = \prod_{i=1}^{|\mathcal{T}|} \frac{\exp\left(\beta \log \frac{\pi^*(\tau^{\sigma(i)}|x)}{\pi_{\text{old}}(\tau^{\sigma(i)}|x)}\right)}{\sum_{j=i}^{|\mathcal{T}|} \exp\left(\beta \log \frac{\pi^*(\tau^{\sigma(j)}|x)}{\pi_{\text{old}}(\tau^{\sigma(j)}|x)}\right)}$$

$$= P_{\pi^*}(\sigma|\mathcal{T}, x)$$

Thus, if the policies are identical, then their induced PL ranking models are also identical.

**Reverse Direction.** Suppose $P_{\pi_\theta}(\sigma \mid \mathcal{T}, x) = P_{\pi^*}(\sigma \mid \mathcal{T}, x)$ for possible permutations $\sigma$, finite trajectory sets $\mathcal{T}$, and queries $x$ (with $\beta > 0$ and $\pi_{\text{old}}(\cdot \mid x) > 0$ on the support). First, define the scores

$$s_\pi(\tau \mid x) := \beta \log \frac{\pi(\tau \mid x)}{\pi_{\text{old}}(\tau \mid x)}.$$

Next, consider any two trajectories $\tau, \tau'$ and the two-trajectory collection $\mathcal{T} = \{\tau, \tau'\}$. The PL probabilities for the two possible permutations satisfy

$$P_{\pi_\theta}\big((\tau, \tau') \mid \mathcal{T}, x\big) = P_{\pi^*}\big((\tau, \tau') \mid \mathcal{T}, x\big), \quad P_{\pi_\theta}\big((\tau', \tau) \mid \mathcal{T}, x\big) = P_{\pi^*}\big((\tau', \tau) \mid \mathcal{T}, x\big).$$

By the PL definition on a two-item set $\mathcal{T} = \{\tau, \tau'\}$, the probability of placing $\tau$ first under $\pi$ will be

$$P_\pi\big((\tau, \tau') \mid \mathcal{T}, x\big) = \frac{e^{s_\pi(\tau \mid x)}}{e^{s_\pi(\tau \mid x)} + e^{s_\pi(\tau' \mid x)}} = \frac{1}{1 + \exp\big(s_\pi(\tau' \mid x) - s_\pi(\tau \mid x)\big)}.$$

Hence equality of the two top-1 probabilities for $\pi_\theta$ and $\pi^*$ implies equality

$$\frac{P_{\pi_\theta}((\tau, \tau') \mid \mathcal{T}, x)}{P_{\pi_\theta}((\tau', \tau) \mid \mathcal{T}, x)} = \frac{P_{\pi^*}((\tau, \tau') \mid \mathcal{T}, x)}{P_{\pi^*}((\tau', \tau) \mid \mathcal{T}, x)} \quad \Longleftrightarrow \quad e^{s_{\pi_\theta}(\tau \mid x) - s_{\pi_\theta}(\tau' \mid x)} = e^{s_{\pi^*}(\tau \mid x) - s_{\pi^*}(\tau' \mid x)}.$$

Taking logarithms yields

$$s_{\pi_\theta}(\tau \mid x) - s_{\pi_\theta}(\tau' \mid x) = s_{\pi^*}(\tau \mid x) - s_{\pi^*}(\tau' \mid x).$$

Fix an arbitrary reference $\tau_0$ and set $\tau' = \tau_0$. The above identity then gives, for every $\tau$,

$$s_{\pi_\theta}(\tau \mid x) - s_{\pi_\theta}(\tau_0 \mid x) = s_{\pi^*}(\tau \mid x) - s_{\pi^*}(\tau_0 \mid x).$$

Based on the invariance property of the exponential scoring and ranking mechanism (Lemma B.2), this is equivalent to the existence of a constant $C(x) := s_{\pi_\theta}(\tau_0 \mid x) - s_{\pi^*}(\tau_0 \mid x)$ (independent of $\tau$) such that

$$s_{\pi_\theta}(\tau \mid x) = s_{\pi^*}(\tau \mid x) + C(x), \quad \forall \tau.$$

Equivalently,

$$\beta \log \frac{\pi_\theta(\tau \mid x)}{\pi_{\text{old}}(\tau \mid x)} = \beta \log \frac{\pi^*(\tau \mid x)}{\pi_{\text{old}}(\tau \mid x)} + C(x),$$

so dividing by $\beta$ and exponentiating gives

$$\frac{\pi_\theta(\tau \mid x)}{\pi_{\text{old}}(\tau \mid x)} = \frac{\pi^*(\tau \mid x)}{\pi_{\text{old}}(\tau \mid x)} e^{C(x)/\beta} \quad \Longrightarrow \quad \pi_\theta(\tau \mid x) = e^{C(x)/\beta} \pi^*(\tau \mid x).$$

Since both $\pi_\theta$ and $\pi^*$ are probability distributions and will sum to 1 over all possible trajectories, $e^{C(x)/\beta} = 1$, which implies $\pi_\theta(\tau|x) = \pi^*(\tau|x)$. Therefore, if the induced PL ranking models are identical, then the underlying policy models will also be identical. Combining the results from both directions will complete the proof.

$\square$

## B.2. Proof of Lemma B.2

In this section, we provide the proof for Lemma B.2 previously used in the proof of Theorem B.1.

---

**Lemma B.2** (Softmax Shift-invariance Property). *Let $z, w \in \mathbb{R}^d$, $d \geq 2$ be two vectors of the same dimension. The softmax outputs are identical, softmax$(z) =$ softmax$(w)$, if and only if there exists a scalar constant $C \in \mathbb{R}$ such that the inputs differ by a constant shift, i.e., $w = z + C \cdot \mathbf{1}$, where $\mathbf{1}$ is the vector of ones. The softmax function is defined component-wise as softmax$(x)_i = \frac{\exp(x_i)}{\sum_{j=1}^{K} \exp(x_j)}$ for a position $i$.*

---

*Proof.* (**Forward** $\Rightarrow$) Suppose vector $w = z + C \cdot \mathbf{1}$ for some constant $C$. Then, for any component $i$:

$$\text{softmax}(w)_i = \frac{\exp(w_i)}{\sum_{j=1}^{K} \exp(w_j)} = \frac{\exp(z_i + C)}{\sum_{j=1}^{K} \exp(z_j + C)}$$

$$= \frac{\exp(z_i)e^C}{e^C \sum_{j=1}^{K} \exp(z_j)} = \frac{\exp(z_i)}{\sum_{j=1}^{K} \exp(z_j)} = \text{softmax}(z)_i.$$

Since this holds for all $i$, softmax$(w) =$ softmax$(z)$.

(**Backward** $\Leftarrow$) Suppose softmax$(z) =$ softmax$(w)$. Let $S_z = \sum_j \exp(z_j)$ and $S_w = \sum_j \exp(w_j)$. This condition implies $\frac{\exp(z_i)}{S_z} = \frac{\exp(w_i)}{S_w}$ for all $i$. Since $S_z, S_w > 0$, we rearrange to get

$$\exp(w_i) = \exp(z_i) \cdot (S_w/S_z).$$

Let the positive constant $K = S_w/S_z$. Taking the natural logarithm yields $w_i = \ln(\exp(z_i)K) = z_i + \ln(K)$. Setting the constant $C = \ln(K)$, we have $w_i = z_i + C$ for all $i$. Thus, we will have $w = z + C \cdot \mathbf{1}$, which completes the proof.

$\square$

## C. Experiment Setups

We evaluate our model on a comprehensive suite of benchmarks to assess its capabilities across various domains. The datasets used are detailed below, categorized by the primary task they are designed to evaluate.

### Mathematical and Scientific Reasoning

- **AIME24 and AIME25** (Maxwell-Jia, 2024; math ai, 2025): The American Invitational Mathematics Examination (AIME) datasets, specifically from the years 2024 and 2025, consist of challenging mathematical problems from this prestigious high school competition. These datasets are used to evaluate the mathematical reasoning and problem-solving abilities of large language models. The problems cover various domains like algebra, geometry, and number theory and require multi-step reasoning.

- **GPQA / GPQA-Diamond** (Rein et al., 2024): GPQA is a benchmark of 448 graduate-level multiple-choice questions in biology, chemistry, and physics, crafted by domain experts and validated to be "Google-proof." Expert accuracy is 65% (or 74% after correction), while skilled non-experts (with unlimited web access) achieve 34%. The "Diamond" subset is often used to select the hardest subset of these questions.

### Search-Based Reasoning

- **HotpotQA** (Yang et al., 2018): A multi-hop QA dataset built over Wikipedia, containing 113k questions. Each question often spans multiple documents, and supporting fact annotations are provided to encourage explainable reasoning.

- **2WikiMultiHopQA (2Wiki)** (Ho et al., 2020): Similar to HotpotQA, this is a multi-hop question-answering dataset created from Wikipedia. It's designed to evaluate the reasoning steps of a model and includes evidence to support the answers. It has different question types, including comparison, inference, and compositional questions.

- **Bamboogle** (Press et al., 2023): This dataset is a collection of questions that are designed to be difficult for Google's search engine to answer correctly. It is used to test the compositional reasoning abilities of language models, pushing them beyond simple information retrieval.

### Multimodal Understanding

- **MMSearch** (Jiang et al., 2024): This is a benchmark for evaluating the multimodal search performance of large language models. The dataset consists of instances that are not present in the training data of current models, so the correct answer can only be found by searching. It evaluates models on tasks like re-querying, reranking, and summarization in a multimodal context.

---

**Data Example of V-Chart**

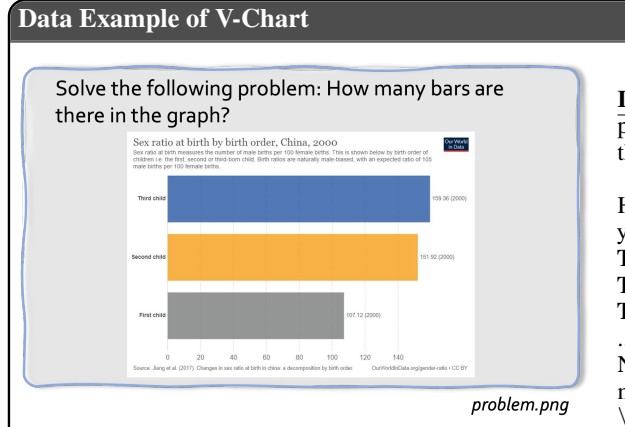

Solve the following problem: How many bars are there in the graph?

*problem.png*

Answer: 3

**Input Prompt:** You are given a problem and a set of tools. Solve the problem step by step by selecting the proper tools to use during your thinking.

Here is the name and usage instructions for the toolset, where you will select the appropriate tools to use.
**Tool 1: Code Interpreter (CI)** ...
**Tool 2: OCR** ...
**Tool 3: SearchAPI** ...
...
Now, given the input image `<problem.png>` about the cryptarithmetic puzzle, solve the problem and put the final answer into `\boxed{}`

---

- **V-Chart**

- **ChartGemma** (Masry et al., 2024): The ChartGemma dataset is used for chart understanding and reasoning. It contains a diverse set of charts and is used to train and evaluate models on their ability to interpret and answer questions about the visual information presented in charts.

- **V-Math**

  - **GSM8K** (Cobbe et al., 2021): This dataset, which stands for "Grade School Math 8K," contains 8,500 high-quality and linguistically diverse grade school math word problems. It is designed to measure the multi-step reasoning capabilities of language models.
  - **AIME24**: Using data from AIME24, we reframe the text data into an image form.

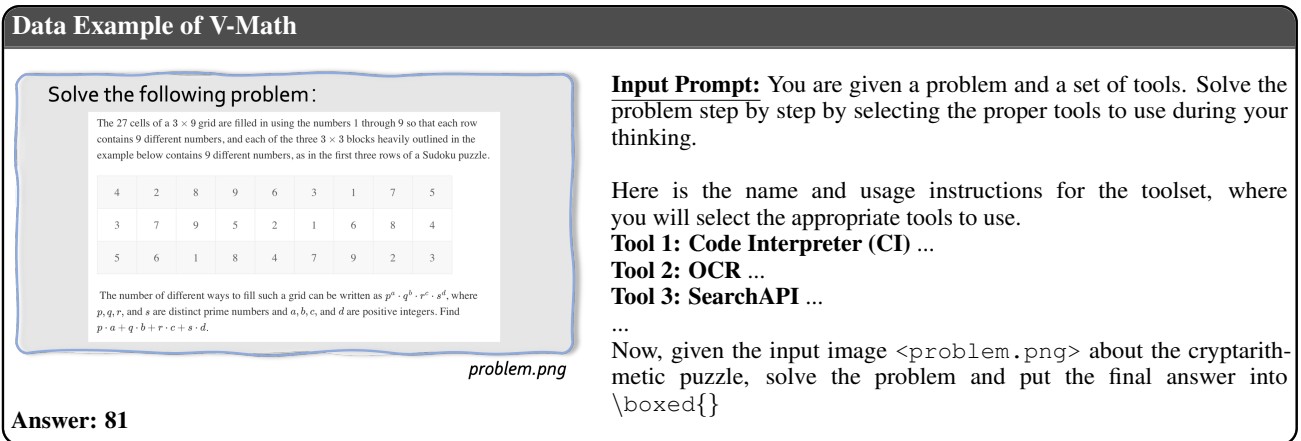

**Data Example of V-Math**

Solve the following problem:

The 27 cells of a 3 × 9 grid are filled in using the numbers 1 through 9 so that each row contains 9 different numbers, and each of the three 3 × 3 blocks heavily outlined in the example below contains 9 different numbers, as in the first three rows of a Sudoku puzzle.

| 4 | 2 | 8 | 9 | 6 | 3 | 1 | 7 | 5 |
| 3 | 7 | 9 | 5 | 2 | 1 | 6 | 8 | 4 |
| 5 | 6 | 1 | 8 | 4 | 7 | 9 | 2 | 3 |

The number of different ways to fill such a grid can be written as $p^a \cdot q^b \cdot r^c \cdot s^d$, where $p, q, r,$ and $s$ are distinct prime numbers and $a, b, c,$ and $d$ are positive integers. Find $p \cdot a + q \cdot b + r \cdot c + s \cdot d$.

*problem.png*

**Answer: 81**

**Input Prompt:** You are given a problem and a set of tools. Solve the problem step by step by selecting the proper tools to use during your thinking.

Here is the name and usage instructions for the toolset, where you will select the appropriate tools to use.
**Tool 1: Code Interpreter (CI)** ...
**Tool 2: OCR** ...
**Tool 3: SearchAPI** ...
...
Now, given the input image `<problem.png>` about the cryptarithmetic puzzle, solve the problem and put the final answer into `\boxed{}`

- **V-Code**

  - **MBPP** (Austin et al., 2021): The "Mostly Basic Python Problems" dataset consists of around 1,000 entry-level Python programming problems. It's used to evaluate the ability of language models to generate code from natural language descriptions.
  - **HumanEval** (Chen et al., 2021): A dataset created by OpenAI, HumanEval consists of 164 handwritten programming problems to evaluate the functional correctness of code generated by language models. The problems are designed to be novel and not easily found on the web, to prevent models from simply recalling existing code.
  - **LiveCodeBench** (Jain et al., 2024): A dynamic benchmark collecting new programming problems from competitive programming platforms (LeetCode, AtCoder, Codeforces), designed to provide "contamination-free" code evaluation by only testing on problems released after model training cutoffs.

**Data Example of V-Code**

Solve the following problem:

Write a function to multiply two lists using map and lambda function.

*problem.png*

**Answer:**

```python
def mul_list(nums1, nums2):
    result = map(lambda x,
    y: x * y, nums1, nums2)
    return list(result)
```

**Input Prompt:** You are given a problem and a set of tools. Solve the problem step by step by selecting the proper tools to use during your thinking.

Here is the name and usage instructions for the toolset, where you will select the appropriate tools to use.
**Tool 1: Code Interpreter (CI)** ...
**Tool 2: OCR** ...
**Tool 3: SearchAPI** ...
...
Now, given the input image `<problem.png>` about the cryptarithmetic puzzle, solve the problem and put the final answer into `\boxed{}`

# D. Additional Related Works

**LLM Agents with Agentic RL.** A prominent line of research on agentic reasoning in LLMs builds upon reinforcement learning to optimize reasoning behaviors directly. GRPO and its variants (Shao et al., 2024; Zheng et al., 2025; Zhao et al., 2025; Li et al., 2025a; Zhang et al., 2025) marked a turning point, proposing a critic-free formulation that estimates group-relative advantages across sampled responses (Liu et al., 2025; Yu et al., 2025a; Lu et al., 2025). Building on these foundations, recent work has explored agentic reasoning with tool use, such as ARTIST (Singh et al., 2025) and rStar2 (Shang et al., 2025), which employ outcome-based RL in settings with noisy or uncertain tool feedback. In parallel, recent research on self-evolving agents (Fang et al., 2025; Yu et al., 2025b; Gao et al., 2025a) shifts the view from static agents toward agents designed for continual adaptation. This line of work emphasizes mechanisms such as continual learning, structural modification, and autonomous self-improvement to extend an agent's reasoning capacity over time.

**Tool Selection in LLM Agents.** Prior work on enabling LLM agents to interact with external tools has explored both retrieval-based selection and tool-generation paradigms (Li et al., 2025b; Qu et al., 2025; Yang et al., 2026; Li et al., 2026). Retrieval-based approaches (Du et al., 2024), employ external hierarchical retrievers to map user queries to candidate tools. These retrievers are trained independently of the LLM agent and operate as standalone indexing systems, resulting in a two-stage pipeline in which the LLM does not directly optimize its internal representations for tool selection. Another line of research focuses on tool generation, including CRAFT (Yuan et al., 2024), ToolMaker (Cai et al., 2023), and CREATOR (Qian et al., 2023), which synthesize new tools or executable code when suitable tools are unavailable. While effective, these methods emphasize tool construction and often incur substantial execution, validation, and debugging overhead. In contrast, AutoTool focuses on the complementary challenge of tool selection by jointly aligning the LLM's internal representations with tool embeddings, enabling efficient and robust generalization to large and evolving toolsets.

# E. Additional Experiments

## E.1. AutoTool vs. Oracle Tool Assignment.

*Table 5.* Comparison with a ground-truth tool assignment baseline where the correct tool is directly provided to the LLM agent before invocation (i.e., no requirements for explicit tool-selection actions).

| **Qwen2.5-VL-7B** | Pre-given (No Tool-Selection) | **AutoTool** |
|---|---|---|
| AIME24 | 46.7 | 46.7 |
| HotpotQA | 35.2 | 33.2 |
| 2Wiki | 38.1 | 36.5 |
| MMSearch | 49.8 | 49.3 |
| V-Math | 43.7 | 44.3 |

To examine the effectiveness of AutoTool's dynamic tool selection abilities, we further conduct experiments to compare AutoTool with a ground-truth tool assignment baseline where the correct tool is directly provided before each invocation. As shown in Table 5, we find that AutoTool achieves performance highly competitive with this "oracle" setup, with only marginal differences across benchmarks (e.g., $49.3$ vs. $49.8$ on MMSearch and $44.3$ vs. $43.7$ on V-Math). This minimized gap demonstrates that AutoTool 's Phase II optimization enables agents to autonomously select tools nearly as effectively as ground-truth assignment, validating the robustness of explicit tool-selection training.

## E.2. Case Study on AutoTool

In Figure 8, we present an example of AutoTool (Qwen3-8B) solving an image-based cryptarithmetic puzzle from V-Math. The agent first invokes OCR to extract textual content from the input image, and then adaptively switches to the Code Interpreter to verify digit assignments and compute the final solution. This case study highlights how AutoTool enables dynamic tool selection for the agent model to seamlessly integrate visual perception with symbolic reasoning.

---

**Case Study on V-Math**

Solve the following problem:

```
      W H I T E
  +   W A T E R
  ---------------
      P I C N I C
```

Each distinct letter represents a different digit.
Find the number represented by the answer **PICNIC**.

*problem.png*

**Input Prompt:** You are given a problem and a set of tools. Solve the problem step by step by selecting the proper tools to use during your thinking.

Here is the name and usage instructions for the toolset, where you will select the appropriate tools to use.
**Tool 1: Code Interpreter (CI)** ...
**Tool 2: OCR** ...
**Tool 3: SearchAPI** ...
...
Now, given the input image `<problem.png>` about the cryptarithmetic puzzle, solve the problem and put the final answer into `\boxed{}`

---

**AutoTool Response:**

Okay, let's tackle this cryptarithmetic puzzle step by step. First, given the input image, I need to extract the information about the puzzle for later calculation.

<**select**> We need to extract the information from the input image and then reason programmatically about the cryptarithmetic problem. OCR is suitable because it converts the visual image into structured text for further automated reasoning. <**tool**>OCR</**tool**> </**select**>

<**ocr**>
{"name": "OCR", "action": "extract_text", "input": "problem.png"}
</**ocr**>
<**interpreter**> Extracted Text: Solve the following problem ... by the answer PICNIC. </**interpreter**>

From the texts, each letter represents a unique digit. We need to find the number represented by PICNIC. First, since adding two 5-digit numbers results in a 6-digit number, the leading digit $P$ must be 1 due to the maximum carryover. So, $P = 1$. Next, we analyze the columns from right to left. The key steps involve checking carryovers and ensuring unique digits. Let's verify possible values for $W$ and $I$ based on the equation $2W + c_4 = 10 + I$.

<**select**> We need to systematically check possible values for $W$ and $I$ based on mathematical constraints. This requires iterating through multiple possibilities and performing calculations. A code execution environment is ideal for this task. <**tool**>Code Interpreter (CI)</**tool**> </**select**>

<**code**>

```python
possible_w_i=[]
for c4 in [0,1]:
  for W in range(2,10):
    I=2*W+c4-10
    if 0<=I<=9 and I!=1:
        possible_w_i.append((W,I,c4))
print(possible_w_i)
```

</**code**>
<**interpreter**>
$[(5, 0, 0), (6, 2, 0), (7, 4, 0), (8, 6, 0), (9, 8, 0), (6, 3, 1), (7, 5, 1), (8, 7, 1), (9, 9, 1)]$
</**interpreter**>

From the output, valid $(W, I)$ pairs are filtered. We then explore each case. After several attempts, we find a valid solution when $W = 8$, $I = 6$, $c_4 = 0$, and other constraints align. Thus, WHITE is 83,642 and WATER is 85,427. Adding together, the PICNIC is 169,069. All digits are unique and satisfy the constraints. Thus, the answer is **\boxed{169069}**.

*Figure 8.* Case Study of AutoTool on V-Math.

