# OpenReview forum: "AutoTool: Dynamic Tool Selection and Integration for Agentic Reasoning"
_ICML.cc/2026/Conference — ICML 2026 regular_

### Official Review · Reviewer_aF7B · 2026-03-09

**Soundness:** 3
**Presentation:** 3
**Significance:** 3
**Originality:** 2
**Overall Recommendation:** 5
**Confidence:** 4

**Summary:**

This paper studies the problem of dynamic tool selection for LLM agents and proposes a training framework called *AutoTool*. The method has two training stages: First, it stabilizes reasoning trajectories with SFT and RL. Second, it introduces a tool selection optimization stage with PL-ranking. To bridge PL ranking with policy optimization, the paper proves that matching the PL ranking distribution induced by the optimal policy is equivalent to learning the optimal rollout policy. The paper also builds a large dataset that contains many tasks and seen/unseen tools. Experiments on several benchmarks show that AutoTool allows the model to learn more reliable tool-selection behaviors without requiring explicit training on specific tool trajectories. LLM agents trained with AutoTool work better via utilizing unseen tools.

**Compliance With Llm Reviewing Policy:**

Affirmed.

**Final Justification:**

All my concerns have been addressed during the rebuttal phase. Thus, I am willing to raise my score to 5.

**Key Questions For Authors:**

- Could the authors provide more detailed design specifications for the anchor token? In particular, it is unclear how this token is trained. To my knowledge, an LLM cannot generate such a compact token for tool retrieval and selection in a zero-shot manner.
- Could the authors provide more detailed information about the PRM design and robustness analysis?

**Limitations:**

Please refer to the major concerns in "Strengths And Weaknesses".

**Strengths And Weaknesses:**

**Strengths:**

- The proposed *AutoTool* framework improves LLM agents' dynamic tool selection capability by introducing an explicit anchor embedding to retrieve relevant tools and a PL-ranking guided training method for tool selection. This design helps the LLM agent better identify and choose appropriate tools even with unseen tools.
- Section 2.3 (Phase II) provides a detailed theoretical analysis based on PL-ranking. The paper clearly proves that the original permutations optimization problem can be transformed into a distribution optimization problem, which makes the training objective more tractable.
- The experiments are comprehensive and directly support the main contributions of the paper, especially in dynamic tool selection and generalization to unseen tools.
- The paper is well written and easy to follow. The introduction clearly explains the motivation of the work, and the section titles and theoretical analysis help readers understand the method step by step.


**Weaknesses:**

***Major Concerns:***

1. Section 2.2 describes the tool embedding as "frozen and precomputed" based on the model's internal state. However, the model's parameters ``\theta`` are updated during Phase I (SFT+RL) and Phase II. This may cause a distribution shift between the query and tool embedding spaces.
2. In Table 2, SFT,GRPO and ARPO baselines seems not use the same embedding-anchored selection mechanism as AutoTool. The performance gains cannot be solely attributed to the Phase II PL-ranking optimization.
3. The reward function ``R_{tool}`` (Eq. 6) relies partially on a PRM to evaluate the quality of tool selection rationales. As the paper does not provide any details about the PRM design, it is unclear if they could provide reliable supervision for tool selection rationales.

***Minor Concerns:***

1. Figure 5’s legend does not align with the bars in the chart: the number of legend entries (5) does not match the number of bars (4), and the colors are also inconsistent.
2. Typo errors:
   - Table 2 Line 333 benchmark names 'GQPA' -> 'GPQA' and Bamboogl -> 'Bamboogle'

---

> ### Author Rebuttal · Authors · 2026-03-31
>
> Dear Reviewer aF7B,
>
> Thank you very much for your insightful feedback and for acknowledging the strengths of our work! We address each of your concerns and questions below.
>
> ---
> ## **[Major Concerns 1] Clarify Tool Embedding Design and Potential Distribution Shift**
> We apologize for the confusion here. We explain our design clearly below.
>
> - **Details on Tool-Embeddings Design:** In our design, tool embeddings are computed from the policy model after each training stage, then kept frozen during the subsequent inference time, **rather than being fixed once from the initial model throughout all training**.
> - **Ablation Analyses:** We carefully analyze the tool embeddings after each phase, and observe **<5.3% little distribution shift** between query and tool embedding spaces.
> - **Meaning of "frozen and precomputed":** In the original draft, our intention of this phrase was to indicate: During inference, tool embeddings are computed once and then kept fixed for both seen and unseen tool selection.
>
> We thank the reviewer for pointing this out. And we have clarified embedding details in `Section 2.2` to avoid confusion.
>
> ---
> ## **[Major Concerns 2] Performance Gains on Phase II & Selection Mechanism**
>
> Thanks for this insightful question! Indeed, AutoTool differs from the mentioned baselines in more than just the Phase II optimization. Below, we (i) first clarify the role of each component, (ii) then include **a new ablation study** to better assess each individual contribution.
>
> **(i) Complementary Roles**\
> Intuitively, the embedding-anchored selection mechanism improves tool-selection generalization, especially to broader and unseen tool pools. Building on this, Phase II PL-Ranking further strengthens this advantage by better aligning query representations with the correct tools during training.
>
>
> **(ii) New Controled Study**\
> To isolate the contribution of Phase II, we equip baselines with the same embedding-anchored mechanism and compare with AutoTool.
>
>
> |Method (Qwen3-8B)|HotpotQA|Bamboogle|V-Math|
> |-|-|-|-|
> |SFT*|34.0|38.6|43.9|
> |GRPO*|38.6|53.1|51.6|
> |ARPO*|43.2|53.7|51.8|
> |**AutoTool**|**45.1**|**56.8**|**53.0**|
>
> **represents using embedding-anchored selection.*
>
> **Key Findings:** Under the same selection mechanism, AutoTool outperforms the baselines across various tasks, while some baselines remain flat or degrade due to poor adaptation. **This demonstrates that Phase II PL-ranking provides additional benefits besides the tool-selection mechanism.**
>
> ---
> ## **[Major Concerns 3 & Q.2] Specify PRM Design with Robustness Analysis**
>
> Thanks for this detailed question. Below, we first (i) provide more detailed specifications of our PRM design, and then (ii) present a new robustness analysis to demonstrate that the PRM provides reliable supervision for tool selection.
>
> **(i) Details on PRM Design**\
> In our implementation, we use an ensemble of step verifiers to provide robust supervision for tool-selection rationales. For math and science tasks, we use `Math-Shepherd-PRM-7B` and `Qwen2.5-Math-PRM-7B`. For search and web tasks, we use `Web-Shepherd-8B`. We also incorporate `DeepSeek-R1` as a generative step verifier to provide an auxiliary judgment signal across domains. The final step-level reward is obtained by aggregating these verifier outputs through majority voting.
>
> We kindly note that, as AutoTool focuses on the overall dynamic tool-selection pipeline, the framework is naturally compatible with additional PRMs or other reward modeling methods.
>
> **(ii) New Robustness Analysis**\
> We further conduct a robustness analysis to assess the reliability of the PRM-based supervision.
>
> **Setups:** We compare the following variants across three science and search tasks.
>
> - **Acc-only** where the tool-selection reward depends only on final answer correctness, without any PRM-based step supervision.
> - **AutoTool (Ours)** which uses the full reward in `Eq.(6)`, combining PRM-based step supervision with outcome accuracy.
>
> **Results:**
> |Method (Qwen3-8B)|GPQA|HotpotQA|Bamboogle|
> |-|-|-|-|
> |Acc-only|71.8|43.6|55.3|
> |**AutoTool (Ours)**|**73.7**|**45.1**|**56.8**|
>
> **Key Findings:** From the results, the performance gains of AutoTool suggest that the PRM design provides meaningful step-level supervision for tool-selection rationales.
>
>
> We thank the reviewer for this detailed question. We have included the specific PRM design and corresponding robustness analyses in the paper for clarification.
>
> ---
> ## **[Minor Concerns 1 & 2]**
> Thank you for pointing these out! We have revised our writing to
> - Align the chart and legend in Figure 5 by using consistent colors and matching bar counts
> - Correct the typos in the two benchmark names.
>
> ---
> ## **[Q.1] Design Specifications for the Anchor Token**
> Thanks for the question!
>
> In our implementation, the anchor token ("<tool_anchor>") is added as a dedicated special token in the tokenizer vocabulary, and the model is trained to generate it at the tool-selection step.

---

> > ### Author Rebuttal · Reviewer_aF7B · 2026-04-03
> >
> > Thanks for your detailed rebuttal. However, I still have several concerns. If these new concerns can be addressed, I am willing to further raise my score.
> >
> > Concern 1: From Rebuttal [Q.1] - Insufficient explanation of the anchor token mechanism.
> >
> > Maybe my last-round review caused some confusion. I would like to clarify my concern again: I want to know how AutoTool enable LLMs to produce different compact anchor tokens for retrieval. To my knowledge, the anchor token should correspond to different embeddings that can be used for retrieval. The response to Rebuttal [Q.1] clarifies that a special token is added into the vocabulary.  Therefore, it seems that the anchor token is static. If the LLM only generates a static anchor token, I do not think it can be used for retrieval.
> >
> > Concern 2: From Rebuttal [Major Concern 2] - Insufficient ablation study
> >
> > Thanks for your additional controlled experiments. The PL Ranking optimization outperforms traditional training algorithms like GRPO and SFT. However, since the authors only provide the selection ratio between seen and unseen tools (*Figure 4*) and the OOD performance (*Table 3*), without any comparison to a more simple and intuitive method (e.g., using the compressed rationale embedding for retrieval), it is still unclear how much the anchor token retrieval mechanism actually improves unseen tool-use performance. From this point of view, I strongly suggest that the authors carry out a small-scale ablation experiment to analyse the effectiveness of the anchor token mechanism.

---

> > > ### Author Response · Authors · 2026-04-03
> > >
> > > Dear Reviewer aF7B,
> > >
> > > Thank you very much for your timely reply and detailed comments! We address each of your remaining concerns below.
> > >
> > > ---
> > > ## **[Concern 1] More Clear and Detailed Explanations of the Anchor Token Mechanism**
> > >
> > > Thank you for this detailed follow-up question! We are happy to clarify the anchor-based tool-selection mechanism more clearly and precisely below.
> > >
> > > > *"To my knowledge, the anchor token should correspond to different embeddings that can be used for retrieval ... If the LLM only generates a static anchor token, I do not think it can be used for retrieval."*
> > >
> > > We fully agree with the reviewer's point here. If retrieval were based only on a fixed, discrete special token, then it would indeed be insufficient for dynamic tool retrieval. That said, we apologize for our previous unclear writing and would like to kindly clarify that AutoTool *does not perform tool selection in this manner*.
> > >
> > > In our method, the anchor token is a **shared special marker** that indicates the tool-selection position, but **the actual retrieval signal is from the dynamic hidden-state representations produced at that position**.
> > >
> > > - **During training**, the model learns to output a sequence of hidden representations (at this anchor position) align with the appropriate target tool embeddings **under different reasoning contexts**. Because the preceding context varies across questions and intermediate trajectories, **the hidden state of the same anchor token is learned to be dynamic**, even though the token ID itself is fixed.
> > > - **During inference**, we therefore **do not directly use the static semantics of the special token itself for retrieval**; instead, we use its context-dependent hidden representation to retrieve the most relevant tool in the embedding space.
> > >
> > > In other words, the anchor token is a **retrieval slot** (similar to a *prefix placeholder token*), not a tool identifier: it specifies where the model should produce a tool-query representation, while **the inner dynamic hidden states at that position determine which tool is selected**. This is also why our method can generalize beyond memorized tool names and select unseen tools via representation alignment rather than exact identifier generation.
> > >
> > > We thank the reviewer for this helpful question! We have carefully refined our writing in `Section 2.2 (Embedding-Anchored Tool Selection)` to (i) clarify the description of the anchor token and (ii) provide more precise details about the selection mechanism.
> > >
> > > ---
> > > ## **[Concern 2] Add New Ablation Study on Anchor Token Mechanism**
> > > Thank you for this great suggestion on the ablation! Below, we conduct **a new ablation study to directly quantify and analyze the effectiveness of the anchor-token mechanism**.
> > >
> > > **Experiment Setups:**\
> > > We compare AutoTool’s anchor-token mechanism with both **an embedding-based variant suggested by the reviewer** and **a generation-based variant**, as detailed below:
> > > - **AutoTool**: Our anchor-token-based tool selection mechanism.
> > > - **Variant 1 (Rationale Embedding-based)**: As suggested by the reviewer, we use the compressed tool-selection rationale embeddings as retrieval signals to match against candidate tools and select the most relevant one.
> > > - **Variant 2 (Generation-based)**: We further include a generation-based variant in which the model reasons over the task and toolset, then directly generates the tool name as its selection.
> > >
> > > We report the downstream performance across different domains and the average tool-selection hit rate for a more thorough assessment.
> > >
> > > **Results:**
> > > |Method|GPQA-D|HotpotQA|Avg. Hit Rate|
> > > |-|-|-|-|
> > > |Variant 1 (Rationale Embedding-based)|71.6|43.5|74.3|
> > > |Variant 2 (Generation-based)|69.2|41.8|58.1|
> > > |**AutoTool**|**73.7**|**45.1**|**83.9**|
> > >
> > > **Key Findings:**
> > > - **Compared with Variant 2**, direct tool-name generation without embedding-based retrieval can't generalize effectively to broader candidate toolsets, especially for unseen tools.
> > > - **Compared with Variant 1**, using the full rationale embeddings as the retrieval signal introduces more irrelevant noise and less fine-grained information, whereas AutoTool’s carefully trained anchor embedding provides a more precise representation for tool matching.
> > >
> > > Overall, the new analyses show that AutoTool’s anchor-token mechanism provides a more effective way to achieve precise and generalizable tool selection, thereby leading to stronger downstream tool-use performance.
> > >
> > > We hope this new ablation study helps the reviewer better understand the effectiveness of AutoTool’s anchor-token mechanism. We have also incorporated this analysis to provide a clearer validation of our method.

---

### Official Review · Reviewer_2tnA · 2026-03-11

**Soundness:** 3
**Presentation:** 4
**Significance:** 2
**Originality:** 3
**Overall Recommendation:** 4
**Confidence:** 4

**Summary:**

This paper proposes a framework named AutoTool to improve LLM agents' tool-selection capabilities, especially on unseen tools. AutoTool trains the agent policy with two phases: (1) SFT+RL to improve agents' reasoning capabilities (2) PL Ranking improves agents' tool-selection capabilities. Experiments on several tasks show that AutoTool outperforms baselines and existing tool-enhanced methods.

**Compliance With Llm Reviewing Policy:**

Affirmed.

**Final Justification:**

My final recommendation is weak accept.

Strengths
1. The paper is well written and easy to follow.
2. The authors construct a 200k-scale tool-use dataset.
3. AutoTool contains a stage specially designed for tool selection.
4. Extensive experiments are performed to demonstrate the effectiveness of AutoTool.

The rebuttal clarifies the toolset split setting and provides an analysis on the impact of tool-selection rationales. My main concerns are addressed. So I raise my rating from 3 to 4.

**Key Questions For Authors:**

See the weaknesses above. I will raise my rating if my concerns are resolved.

**Limitations:**

Yes

**Strengths And Weaknesses:**

Strengths
1. The paper is well written and easy to follow.
2. The authors construct a 200k-scale tool-use dataset.
3. AutoTool contains a stage specially designed for tool selection.
4. Extensive experiments are performed to demonstrate the effectiveness of AutoTool.

Weaknesses
1. The authors criticize existing approaches for assuming a fixed inventory of tools. In Section 3, the 'Supporting Unseen and Evolving Toolsets' subsection introduces the tools used in different stages. The key design is only a subset of tools is used during the dual-stage training; for the tool selection rational generation, a different subset of tools is used. The author claims that this separation between toolsets enables AutoTool to use unseen tools better at inference. However, I am not convinced by the claims that this design is especially effective for unseen tool use. If compared with the toolset used in the training of the baseline model, many post-training methods using self-collected tools could be considered training on unseen toolsets. To me, AutoTool is still using a fixed inventory of tools; it just has two splits of tools. This design indeed improves the performance on some benchmarks compared with baselines according to the experiment results. Could the authors elaborate more on whether the gains come from the separation between toolsets or simply the high quality of the dataset?
2. In Table 2, AutoTool only shows marginal improvements compared with ARPO. Does this indicate that agentic training methods are the most significant part while the training on the tool-selection rationale only has a small influence?

---

> ### Author Rebuttal · Authors · 2026-03-31
>
> Dear Reviewer 2tnA,
>
> Thanks for your thoughtful feedback! We address each of your questions below.
>
> ---
> ## **[W.1] Toolset Split Setting in Unseen Tool Generalization**
> Thank you for this insightful question! Below, we (i) first explain the role of the seen/unseen tool split in our setup, (ii) then provide additional analyses to elaborate on where AutoTool’s gains come from.
>
> **(i) Clarification on Tool Split Design**
>
> **The seen/unseen split is introduced to separate training from inference, not to separate rationale generation from dual-stage training.**
>
> In `Section 3: Supporting Unseen and Evolving Toolsets`, we don't use one tool-subset for selection rationale generation and another subset for dual-stage training. Both the selection rationale data and other reasoning trajectories are drawn from the same 460 seen tools during dual-stage training; the remaining 886 tools are strictly held out for inference only.
>
> Our **design intention** is to provide a clean, leakage-free evaluation environment, rather than a mechanism that directly improves AutoTool’s ability to use unseen tools. By doing so, we can faithfully assess whether AutoTool's tool-selection ability can generalize to the expanded unseen tools during inference.
>
> We apologize for the confusion here and have revised the presentation to explicitly clarify the motivation behind the seen/unseen tool split and its role in our setup.
>
> **(ii) New Analyses for Elaboration**
>
> To better answer reviewer’s question, we conduct new experiments to show AutoTool’s performance gains are not due to the held-out split, but **arise from both selection rationale curation and our dual-stage training paradigm refinement**.
>
> **[Analysis 1] Ablation on Selection Rationale**
>
> We compare AutoTool (Qwen3-8B) with a variant trained without tool-selection rationales to evaluate the efficacy of our curated data.
> |Method|GPQA-D|HotpotQA|V-Chart|
> |-|-|-|-|
> |AutoTool w/o Rationale|71.4|42.8|22.3|
> |**AutoTool**|**73.7**|**45.1**|**24.7**|
>
> **Key Findings:**  Our curated tool-selection rationales indeed improve AutoTool’s generalization to unseen tools.
>
> **[Analysis 2] Comparison with Baseline under Same Tools & Data**
>
> We further compare AutoTool with a strong baseline ReTool, under a controlled setting in which both methods are trained on the same data and toolset.
>
> |Method|AIME25|GPQA-D|HotpotQA|
> |-|-|-|-|
> |ReTool|49.6|70.3|35.9|
> |**AutoTool**|**51.2**|**73.7**|**45.1**|
>
> **Key Findings:** Under the same high-quality training data, AutoTool improves over the baseline, highlighting the effectiveness of our Phase-II PL-ranking design in generalizing to unseen tools.
>
> We have incorporated new analyses in our paper to better elaborate on AutoTool’s gains.
>
> ---
> ## **[W.2] Analyzing the Impact of Tool-Selection Rationales (Comparison with ARPO)**
> Thanks for the question! We (i) first clarify that the modest gain over ARPO is mainly due to differences in the experimental setups, (ii) then provide a more controlled study that explicitly isolates the effect of training with tool-selection rationales.
>
> **(i) Explain why gains in Table 2 appear modest**
>
> `Table 2` compares ARPO and AutoTool as **two different complete training pipelines rather than a strictly controlled ablation:**
> - For AutoTool setup in Table 2, we apply standard GRPO as the Phase-I RL algorithm. This design is intended to enable a cleaner comparison between the full dual-phase pipeline and variants with only Phase-I training (i.e., rows of "SFT" and "GRPO").
> - For the ARPO baseline, we use same training data and overall compute budget, **but we didn't directly apply ARPO within AutoTool**. Our intention is to compare ARPO and AutoTool as two different end-to-end training pipelines.
>
> Thus, using the default GRPO rather than the stronger ARPO in Phase I of AutoTool helps explain the relatively modest gains, since ARPO generally suits agentic tool use better than standard GRPO.
>
> **(ii) Add Controlled Study to Access Tool-Selection Rationales**
>
> >Does this indicate that agentic training are most significant part, while training on selection rationale only has a small influence?
>
> To better answer this, we add a new control study: using ARPO as the Phase-I RL, then continue training with Phase II. This isolates and assesses whether "training on tool-selection rationale" (on Phase II) is important.
> |Method (Qwen3-8B)|GPQA-D|HotpotQA|2Wiki|V-Math|
> |-|-|-|-|-|
> |Phase-I (ARPO)|72.8|43.6|48.1|52.6|
> |**Phase-I (ARPO) + Phase II (tool-selection refinement)**|**74.5**|**47.7**|**51.3**|**55.4**|
>
> **Key Findings:** While Phase I ARPO provides the performance foundation, Phase II training on tool-selection rationales brings additional gains. This suggests training on tool-selection refinement is equally important to the overall AutoTool pipeline.
>
> We have revised `Section 4` to clarify the implementation details in Table 2, and added the control study to directly evaluate the impact of tool-selection rationales during training.

---

> > ### Author Rebuttal · Reviewer_2tnA · 2026-04-02
> >
> > Thank you for your responses. My concerns have been addressed. I raise my overall score to 4.

---

> > > ### Author Response · Authors · 2026-04-02
> > >
> > > Dear Reviewer 2tnA,
> > >
> > > Thank you very much for your positive rating on our submission! We are glad that our clarifications fully addressed your questions, and we sincerely appreciate your time, effort, and support of our work!
> > >
> > > Best regards,
> > >
> > > Authors of AutoTool

---

### Official Review · Reviewer_mHAe · 2026-03-12

**Soundness:** 2
**Presentation:** 3
**Significance:** 2
**Originality:** 3
**Overall Recommendation:** 4
**Confidence:** 4

**Summary:**

This paper introduces AutoTool, which is designed to improve tool selection in LLM agents. The authors introduce a two-phase training pipeline, including an SFT+RL stage to enhance the model's CoT capability and a second phase to improve the agent's tool selection ability. Besides, the paper models the tool selection as a Plackett-Luce ranking problem to capture the relative ordering relationships among candidate tools. Experiments on various datasets (i.e., math, search, etc.) demonstrate that the proposed method outperforms existing agentic RL approaches.

**Compliance With Llm Reviewing Policy:**

Affirmed.

**Key Questions For Authors:**

Please refer to the paper's weakness. I suggest the author conduct experiments on benchmarks (need more tools) and I will reconsider the rating if the further experiments could verify the advantages of AutoTool.

**Limitations:**

yes

**Strengths And Weaknesses:**

**Strengths:**
1. The embedding-anchored mechanism is scalable. By decoupling tool descriptions from the prompt and using pre-computed embeddings, it naturally supports a massive toolset and zero-shot generalization to new tools.
2. The connection established between the optimal RL policy and the Plackett-Luce ranking distribution provides a deeper understanding of the training objective.

**Weaknesses**:
1. Limited Tool Diversity in Current Benchmarks: While the paper claims advantages in handling massive toolsets, the current evaluation benchmarks may not fully reflect this capability. There is a lack of statistical analysis regarding the types of tools used in the evaluation benchmarks. For example, in typical Math benchmarks, the agent predominantly calls a Python/Code interpreter; in standard search QA tasks, it almost relies on Search tools. This limited tool-use variance per query may fails to demonstrate the true superiority of AutoTool in complex routing scenarios. The authors may conduct further experiments on benchmarks that require using more tools, such as GAIA and GAIA2.
2. The experiments demonstrate zero-shot generalization to unseen tools, but the paper lacks a detailed definition and analysis of what constitutes these tools. It is crucial to understand the relationship between the seen and unseen tools in the embedding space. If the unseen tools are semantically very similar to the seen ones (i.e., the distance between their embeddings is extremely close), the generalization claim might be overstated.
3. The paper needs to clarify the details of its two-stage training paradigm. (1) Could optimizing purely for tool selection accuracy lead to a degradation in the final task completion rate (outcome success)? (2) Is the reward in the first stage outcome-based?
4. I wonder whether the agentic RL baselines were retrained or fine-tuned on the exact same toolset and trajectories used for AutoTool. If they were not retrained, their performance degradation might simply be attributed to a lack of  certain specific tools (e.g., multimodal tools) rather than an architectural inferiority in tool selection.

---

> ### Author Rebuttal · Authors · 2026-03-31
>
> Dear Reviewer mHAe,
>
> Thank you very much for your constructive feedback! We address each of your concerns below.
>
> ---
> ## **[W.1 & `Key Questions For Authors`] Extend Tool Diversity to Benchmarks (Need More Tools)**
> Thank you for this insightful suggestion! Below, we (i) first clarify how our current work involves multi-turn tool use, (ii) then conduct **new experiments on more challenging benchmarks.**
>
> **(i) Analyses on Existing Multi-turn Tasks**\
> In our paper, we have 3 test sets (V-Chart, V-Math, V-Code) that require multi-turn different tool use. Each question generally requires AutoTool to invoke tools spanning a large portion of the toolset, as analyzed:
> |Tasks|Avg. Tool Type per Query|Total Coverage of Toolset|
> |-|-|-|
> |V-Chart|4.3|83.6%|
> |V-Code|4.1|71.2%|
> |V-Math|3.6|69.3%|
>
> **(ii) New Assessment on More Complex Scenarios**
>
> We follow the reviewer's suggestion and add two more challenging agentic benchmarks, GAIA [1] and Humanity's Last Exam (HLE) [2]. Each task requires intensive multi-round and different tool use ($\ge 4$).
> |Methods|GAIA|HLE|Avg.|
> |-|-|-|-|
> |Qwen2.5-VL-72B-Instruct|18.5|2.2|10.4|
> |DeepSeek-R1-Distill-Qwen-14B|27.1|6.4|16.8|
> |ARPO (Qwen3-8B)|31.2|8.6|19.9|
> |**AutoTool (Qwen3-8B)**|**34.8**|**9.3**|**22.1**|
>
> **Key Findings:** The new evaluations strengthen our claim that AutoTool’s dynamic tool selection can adapt to more challenging settings with richer tool compositions. We have included the new experiments and analyses in our paper.
>
> ---
> ## **[W.2] Similarity between Seen and Unseen Tools**
> Thanks for the question! Below, we (i) first elaborate more on the definition of tool-split, (ii) then provide **a detailed similarity analysis** between seen and unseen tools.
>
> **(i) Details on Current Seen/Unseen Tools**\
> We collect the toolset from over 150 independent sources, such as online APIs, open-source libraries, and multimodal utilities, spanning 126 task types. This diverse and decentralized collection process makes it less likely that the seen tools are similar to unseen ones.
>
> **(ii) New Similarity Analyses**\
> We further quantify the embedding-space cosine similarity between the seen and unseen tools, and compare with another large tool inventory, ToolBench [3].
> |Toolset|Domain|Avg. Cosine Sim.(↓)|
> |-|-|-|
> |ToolBench|Overall|0.427|
> |AutoTool (Seen v.s. Unseen)|Math&Science|0.364|
> |AutoTool (Seen v.s. Unseen)|Search|0.412|
> |AutoTool (Seen v.s. Unseen)|Visual|0.308|
> |**AutoTool (Seen v.s. Unseen)**|**Overall**|**0.361**|
>
> **Key Findings:** The embedding-space distance between AutoTool’s seen and unseen tools is generally large, with low semantic similarity rather than near-duplicate overlap. The new analyses here demonstrate the validity of AutoTool’s generalization ability.
>
> ---
> ## **[W.3] Details on Two-stage Training Paradigm**
> Thanks for these detailed questions! We provide detailed responses below.
> > (1) Could optimizing purely for tool selection accuracy lead to a degradation in the final task completion rate (outcome success)?
>
> Generally, we didn't observe such a trade-off on final task completion in AutoTool.
>
> In **Phase II reward assignment** (`Eq.(6)`), the step-wise reward combines both tool-selection quality (measured by PRM) and the end-task correctness (measured by Acc$(x, t_k)$). Using this reward signal during Phase II training encourages the policy to account for both the quality of tool selection and the resulting task success.
>
> In addition, **Phase I** optimizes the entire trajectory, rather than only the tool-selection rationales, which helps stabilize the policy model in both tool-call execution and inner-reasoning completion.
>
> **Empirically,** results on Table 2 and Figure 5 show that incorporating Phase II yields an additional 18.6% improvement in tool-selection hit rate and 10.4% performance gains over Phase I alone. This suggests that Phase II training further improves the model through more effective tool-aware optimization.
> > (2) Is the reward in the first stage outcome-based?
>
> In AutoTool, Phase I mainly serves as a preliminary trajectory-stabilization stage. Accordingly, the RL objective used in Phase I is not restricted to a specific reward type and can be instantiated as either process-based or outcome-based, depending on the chosen RL algorithm. In our experimental implementation, we adopt standard GRPO, which uses an outcome-based reward.
>
> ---
> ## **[W.4] Clarify the Baseline Training Setup**
> Thanks for the great question!
>
> In our experiments, agentic RL baselines (e.g., ARPO) **are retrained on the same training data** as AutoTool, including the same toolset and CoT trajectories. We also ensure to match the overall compute budget, such as overall training steps and GPU environment, to ensure a fair and faithful comparison with AutoTool.
>
> ---
> ## **References:**
> [1] GAIA: A Benchmark for General AI Assistants.\
> [2] Humanity's Last Exam.\
> [3] Toolllm: Facilitating LLMs to master 16000+ real-world APIs.

---

> > ### Author Rebuttal · Reviewer_mHAe · 2026-04-03
> >
> > The authors address my concerns. I'll still keep my relative positive score.

---

> > > ### Author Response · Authors · 2026-04-03
> > >
> > > Dear Reviewer mHAe,
> > >
> > > Thank you very much for your positive rating of our paper! We are pleased that our rebuttal has addressed your concerns. We sincerely appreciate your time and valuable feedback!
> > >
> > > Best regards,
> > >
> > > Authors of AutoTool

---

### Official Review · Reviewer_JDvh · 2026-03-24

**Soundness:** 3
**Presentation:** 3
**Significance:** 3
**Originality:** 3
**Overall Recommendation:** 5
**Confidence:** 4

**Summary:**

The authors propose a training framework for LLM agents to support dynamic tool selection from large, evolving toolsets during reasoning. The proposed method uses a dual-phase training pipeline: Phase 1 uses SFT + RL training to stabilize the reasoning trajectories, and Phase 2 refines the tool selection behavior using KL regularized Plackett-Luce ranking objective. In practice, authors use the cross-entropy loss and show that optimizing the policy is equivalent to the PL ranking objective. Interestingly, tool selection is performed in embedding space rather than text space. The policy produces an anchor token and selects tools by searching the nearest-neighbor matching against a frozen, precomputed tool embedding set.
To facilitate the training, the authors curate a dataset with 200k instances and add explicit tool-selection rationales using expert LLMs. The instances are taken from the training sets of existing datasets spanning math, science, search-based QA,c code generation, and multimodal tasks. Experimental validation shows that the trained model generalizes to unseen tools at inference time and outperforms the selected baselines.

**Compliance With Llm Reviewing Policy:**

Affirmed.

**Final Justification:**

Since most of my questions concerned missing information from the paper, I believe the authors have provided most of the important details needed to understand the paper, and these should be included in the final copy of the work. In light of this, I am increasing my overall score to 5, and soundness to 3.

**Key Questions For Authors:**

1. Please describe the credit assignment for the accuracy component in equation 6. Additionally, please provide details on which PRM was used for the training. If the PRM was trained on any of the evaluation sets, then there is also an indirect contamination risk.
2. While Appendix E.2 provides a case study for positive cases, authors should provide an error analysis for at least one of the models. It will be fruitful to know what kind of errors the model makes after training.
3. Please provide the exact breakdown of the 200k training set instances along with their source datasets. This is a major point of weakness, as we cannot judge any contamination risk and thus cannot form any opinions on the results.
4. How is the anchor token used during training? Is it a specialized token that is added to the vocabulary of the model?
5. What is the value of $\gamma$ in eq 4?
6. Eq 3 says that the internal embedding layer is used to get tool representations. Which layer embeddings are used here?
7. What evaluation criteria are used for QA tasks like HotPotQA, 2Wiki, or Bamboogle? Is it an exact match? Please specify that.

**Limitations:**

Authors don't discuss the limitations of their proposed pipeline.

**Strengths And Weaknesses:**

## Strengths
- The problem of dynamic tool selection is well motivated, especially for small models.
- Experiments show consistent cross-domain performance, proving the generalizability of the method across domains.
- Table 3 shows that the proposed model generalizes well to using only unseen tools at inference time.
- Table 2 and Figure 5 show that the two-phased approach works well and beats the standard training pipelines.
- Generally, experimental validation is well done with a few caveats (see weaknesses and questions).
- The paper is well written and easy to follow.

## Weaknesses
1. Authors use a Process Reward Model to assign a step-level reward for each tool choice. However, there are no details provided about this PRM. Moreover, it is not clear how the authors calculate the $Acc(x, t_k)$ for each tool choice. A trajectory contains multiple tool calls, and the final answer depends on all tool calls in the trajectory. How is the credit assignment done here? These details are a crucial part of the work and should be provided.
2. The authors propose to generate an anchor token and use the embeddings from the policy model to select the tool. The authors claim that generating tool names in text would fail for unseen tools, but there is no experimental validation of this claim.
3. Results are reported using a single evaluation and training run. RL-based training is known to exhibit high variance across seeds, and evaluations using LLMs are always conducted using multiple runs. It is important to know the variance of the results under these stochastic settings. This is further compounded by the small test set sizes for some of the benchmarks, such as AIME24/25, where a single problem is worth ~3.3%. Several reported gains fall within this granularity, and without multiple runs, it is impossible to determine whether the improvements are genuine or noisy.
4. One concern is that authors use frozen tool embeddings. I am assuming the tool embeddings are frozen throughout the training process since it is not clear in the paper. If that is the case, then the frozen embeddings can drift out of alignment during training. Authors should discuss this point and, if possible, provide an ablation where they update the tool embeddings during training (after each phase or gradually after each epoch).
5. Claiming balanced performance relative to Qwen2.5-Math-72B on search tasks is not a meaningful comparison. The math model is specifically tuned on math datasets. While it is okay to keep it as a baseline, the claim that the AutoTool model provides better generalization than a math-tuned model is not very informative. Furthermore, a lot of the frontier models are missing from the result. Authors should at least include one frontier LLM (GPT-5, Claude Sonnet, Gemini 3 Flash/Pro) to establish the proper baseline. If the frontier models can already generalize to unseen tools, then the claims of this work and the proposed training pipeline only apply to small models and may not be needed at scale.
   Note that I am not commenting on the contribution of the paper here. Even if the claims are only valid for small models, it is still a valid contribution, but it should be clear from the study.
6. The authors mention that they curate a dataset of 200k instances using existing data sources. However, the training data sources are never mentioned in the paper. Without the training data details, it is impossible to judge if there was any contamination between train/test sets.

---

> ### Author Rebuttal · Authors · 2026-03-31
>
> Dear Reviewer JDvh,
>
> Thanks for your constructive comments! We address each of your question below.
>
> ---
> ## **[W.1 & Q.1] Details on PRM and Eq.6 (Acc)**
> Thanks for the insightful question!
>
> **(i) PRM Details:** We use an ensemble of step verifiers for robust supervision. For math/science, we use Math-Shepherd-PRM-7B and Qwen-Math-PRM-7B. For search, we use Web-Shepherd-8B. We also adopt DeepSeek-R1 to provide auxiliary judgment signals across domains. We ensure none of the PRMs are trained on test sets to avoid contamination.
>
> **(ii) Acc Assignment:** Acc is a binary outcome reward assigned to each tool-selection step independently based on the end-task correctness.
>
> ---
> ## **[W.2] Validate Why Tool Names Failed**
> Thanks for the question! We provide detailed explanations and empirical validations on anchar-based selection below.
>
> (i) In our paper, we already compare with several baselines (ReTool, Toolformer) that generate tool names in text.
>
> (ii) **New Ablation Analyses.** We further compare AutoTool (Qwen3-8B) with a variant that uses direct tool-name generation instead of anchor-based selection.
> |AutoTool|GPQA-D|HotpotQA|
> |-|-|-|
> |Tool-Name|70.4|42.6|
> |**Anchor**|**73.7**|**45.1**|
>
> **Key Findings:** anchor-based selection generalizes more effectively than direct tool-name generation.
>
> ---
> ## **[W.3] Experiment Sensitivity & Multi-Run Evaluation**
> Thanks for the question! In main results of `Table 1`, we indeed report the AIME24/25 over 8 independent runs. We also closely monitor training dynamics via Wandb and find AutoTool’s training trajectory to be overall stable.
>
> **Multiple-Run Validation:** We newly conduct 16 independent runs and compare with strongest baseline ARPO in Table 2. AutoTool achieves a non-trivial average gain of 6.1%.
>
> ---
> ## **[W.4] Clarification on Tool Embeddings**
> We apologize for the confusion. We explain our design clearly.
> - **Design Details:** Tool embeddings are computed from the policy model after each training stage, then kept frozen during the subsequent inference time, **rather than being fixed once from the initial model throughout all training**.
> - **Ablation Analyses:** We carefully analyze tool embeddings after each phase, and observe **<5.3% little distribution shift** between the embedding spaces.
> -  Our intended meaning is that: During inference, tool embeddings are computed once and then kept fixed for both seen and unseen tool selection.
>
> We have clarified the details to avoid confusion.
>
> ---
> ## **[W.5] Compare with Frontier Models**
> Thanks for the suggestion! We provide detailed clarification and new comparisons with frontier models.
>
> **(i) Current Comparison Scope**\
> Our goal in current experiments is to compare AutoTool against both domain-specialized reference models and strong general-purpose LLMs. Accordingly, `Table 1` also includes general models like GPT-4o and QwQ-32B to assess whether AutoTool achieves more balanced performance across domains.
>
> **(ii) New Comparison with Frontier Models**\
> We further train AutoTool on Qwen3-14B and compare with two frontier models, GPT-5 (gpt-5-2025-08-07) and Claude Sonnet 4 (claude-sonnet-4-20250514_16k).
> ||GPQA-D|V-Math|Avg.|
> |-|-|-|-|
> |Claude Sonnet 4|73.8|58.4|66.1|
> |GPT-5|85.0|61.7|73.4|
> |**AutoTool**|82.4|67.2|**74.8**|
>
> **Key Findings:** Frontier models like Claude Sonnet 4 don't always generalize well to unseen tools, whereas AutoTool remains competitive across diverse domains.
>
> ---
> ## **[W.6 & Q.3] Training Data Breakdown**
> We are happy to provide additional training data details!
>
> **Training Set and Composition Breakdown.** Please see table in https://anonymous.4open.science/api/repo/Autotool_Submission-72E1/file/img/data.pdf?v=e5954853
>
> **Avoid Contamination.** We ensure all sources **do not contaminate** any downstream test sets by following the policies:
> - Exclude instances from test benchmarks or their indirect source families from training.
> - All source datasets are taken from their public training splits only.
>
> ---
> ## **Detailed Reply to Additional Questions**
> >[Q.2] Error Analysis on AutoTool
>
> Thanks for the suggestion! We provide error analyses on AutoTool (Qwen3-8B). AutoTool mainly fails on:
> - **Tool Execution Error:** The model selects the correct tool but fails to execute properly due to malformed tool inputs (e.g., invalid code snippet generated by the model).
> - **Tool Result Utilization Error:** AutoTool executes correctly, but doesn't effectively incorporate the returned result into subsequent reasoning.
>
> >[Q.4] Anchor Token
>
> Great question! The anchor ("<tool_anchor>") is a dedicated special token in the tokenizer vocab, and the model is trained to generate it at the tool-selection step.
> >[Q.5] Value of $\gamma$
>
> We performed a hyperparameter study and set $\gamma=1.0$.
> >[Q.6] Embedding Layers for Tool Representation
>
> Tool representations are obtained from the policy model’s input (token) embedding layer.
> >[Q.7] Metric of Search QA Tasks
>
> We use Exact Match as the metric for search tasks.

---

> > ### Author Rebuttal · Reviewer_JDvh · 2026-04-04
> >
> > Since most of my questions concerned missing information from the paper, I believe the authors have provided most of the important details needed to understand the paper, and these should be included in the final copy of the work. In light of this, I am increasing my score to 5.
> >
> > However, I have a few questions and comments (minor):
> >
> > **[W1 & Q1]:** Since the $\text{Acc}(x,t_k)$ is the same for each tool-selection step within a trajectory, eq 6 denotes the average of the PRM reward across the trajectory + 1/0 (based on end-task correctness). Is my understanding correct? If yes, then have you done any ablation by removing the accuracy assignment term?
> >
> > **[W2]:** Thanks for providing the ablation. The reason baselines cannot be considered for this comparison is that the training data and training phases are not comparable between baselines and AutoTool. Please add the ablation to the paper.
> >
> > Thank you for providing detailed answers (in limited space!) to all the questions.

---

> > > ### Author Response · Authors · 2026-04-04
> > >
> > > Dear Reviewer JDvh,
> > >
> > > Thank you very much for your constructive feedback and for the positive rating of our submission! We have ensured to include all important details in the paper for clarity.
> > >
> > > We are also happy to provide additional responses below.
> > >
> > > ---
> > >
> > > ## **[W1 & Q1] Analyses & Ablation on Accuracy Assignment Term**
> > > Thanks for the insightful question! We provide detailed explanations and ablations on the $\text{Acc}$ term below.
> > >
> > > > *"Since the end-task correctness is the same for each tool-selection step within a trajectory, eq 6 denotes the average of the PRM reward across the trajectory + 1/0 (based on end-task correctness). Is my understanding correct?"*
> > >
> > > Yes, your interpretation of our reward assignment is exactly right! The **intuition behind incorporating the end-task correctness reward into each tool-selection step** is to encourage the policy model to account for both the *local quality* of each tool choice and its *global contribution* to successful task completion.
> > >
> > > **Ablation Study on Acc**\
> > > We have indeed conducted ablations on this component before. We elaborate further below by comparing AutoTool with the following variants:
> > > - **AutoTool (Ours)** which uses the full reward in `Eq.(6)`, combining PRM-based step supervision with outcome accuracy.
> > > - **PRM-only** where the tool-selection reward depends only on the PRM supervision without end-task correctness assessment.
> > > - **Acc-only** where the tool-selection reward depends only on final answer correctness.
> > >
> > > |Method (Qwen3-8B)|GPQA|HotpotQA|Bamboogle|
> > > |-|-|-|-|
> > > |Acc-only|71.8|43.6|55.3|
> > > |PRM-only|72.5|44.8|55.7|
> > > |**AutoTool (Ours)**|**73.7**|**45.1**|**56.8**|
> > >
> > > **Key Findings:** PRM rewards provide richer step-level supervision for tool selection, and incorporating both PRM-based and outcome-aware rewards into AutoTool further improves final task completion and overall downstream performance.
> > >
> > > ---
> > >
> > > ## **[W2]: Reply to Additional Comments for Clarity**
> > > Thanks for providing the detailed comments!
> > >
> > > > *"The reason that baselines (e.g., ReTool) cannot be considered for this comparison is that the training data and training phases are not comparable between baselines and AutoTool. Please add the ablation to the paper."*
> > >
> > > We agree with the reviewer's point here. For better clarity and presentation, we have followed the reviewer’s suggestion to:
> > > - explicitly describe the experimental setup used for the comparison between AutoTool and the baselines (e.g., training settings);
> > > - add the new ablation on the anchor-based selection mechanism to the paper.
> > >
> > > ---
> > > We hope our explanations above help fully address the reviewer’s questions. And we sincerely appreciate the reviewer’s time, effort, and support for our work!

---

### Decision · Program_Chairs · 2026-04-30

**Decision:**

Accept (regular)

**Comment:**

The paper introduces AutoTool, a two-phase training framework designed to enhance LLM agents' ability to dynamically select tools from large, evolving sets. The pipeline consists of:
- Phase I: Trajectory stabilization via SFT and RL to establish coherent CoT and tool-use patterns.
- Phase II: Tool-selection refinement using a KL-regularized PL Ranking objective.

A key technical contribution is the embedding-anchored selection mechanism, which uses a dynamic hidden-state representation at a specific "anchor" token to retrieve tools, rather than relying on text-based tool names. The authors also contribute a 200k-scale dataset with explicit tool-selection rationales across 1,000+ tools.

Strengths:
- The method shows strong zero-shot generalization to unseen tools at inference time, outperforming baselines that rely on fixed tool inventories.
- The anchor-based retrieval is more robust than generating tool names, which often fails when toolsets evolve or expand.
- The authors tested the framework across diverse domains (Math, Science, Search QA, and Multimodal reasoning) using Qwen3-8B and Qwen2.5-VL-7B.
- The authors provided critical ablations during the discussion, including a comparison with frontier models (GPT-5, Claude Sonnet 4) and a detailed analysis of the anchor token's dynamic nature.

Weaknesses:
- The Phase II reward relies on an ensemble of PRMs and final outcome rewards. While the authors clarified the credit assignment, this dependency adds complexity to the training pipeline.
- In specific benchmarks, the improvement over strong agentic baselines like ARPO is relatively modest, though consistent.
- While the authors provided a similarity analysis, the effectiveness of the embedding space still largely depends on the quality of the initial model's representations.

While the contribution is somewhat incremental relative to existing RLHF/RL methods, the integration of PL-ranking for tool selection and the successful demonstration of unseen tool generalization make this a solid contribution to the field of agentic LLMs.